# Quantifying the potential of using SMAP soil moisture variability to predict subsurface water dynamics

Aruna Kumar Nayak[1], Xiaoyong Xu[1,2], Steven K. Frey[3,4], Omar Khader[3,5], Andre R. Erler[3], David R. Lapen[6], Hazen A.J. Russell[7], and Edward A. Sudicky[3,4]

[1]Department of Chemical and Physical Sciences, University of Toronto Mississauga, Mississauga, ON, Canada
[2]Department of Geography, Geomatics and Environment, University of Toronto Mississauga, Mississauga, ON, Canada
[3]Aquanty, Waterloo, ON, Canada
[4]Department of Earth and Environmental Sciences, University of Waterloo, Waterloo, ON, Canada
[5]Department of Water and Water Structural Engineering, Zagazig University, Al Sharqia, Egypt
[6]Agriculture and Agri-Food Canada, Ottawa Research and Development Centre, Ottawa, ON, Canada
[7]Natural Resources Canada, Ottawa, ON, Canada

*Correspondence to*: Xiaoyong Xu (xiaoyong.xu@utoronto.ca)

**Abstract.** Advances in satellite Earth observation have opened up new opportunities for a global monitoring of soil moisture (SM) at fine to medium resolution, but satellite remote sensing can only measure the near-surface soil moisture (SSM). As such, it is critically important to examine the potential of satellite SSM measurements to derive the water resource variations in deeper subsurface. This study compares the SSM variability captured by the Soil Moisture Active and Passive (SMAP) satellite and the Soil Water Index (SWI) derived from SMAP SSM with subsurface SM and groundwater (GW) dynamics simulated by a high resolution fully-integrated surface water - groundwater model over an agriculturally-dominated watershed in eastern Canada across two spatial scales, namely SMAP product grid (9 km) and watershed (~4000 km$^2$). SMAP measurements compare well with the hydrologic simulations in terms of SSM variability at both scales. Simulated subsurface SM and GW storage show lagged and smoother characteristics relative to SMAP SSM variability with an optimal delay of ~1 days for the 25–50 cm SM, ~6 days for the 50–100 cm SM, and ~11 days for the GW storage for both scales. Modelled subsurface SM dynamics agree well with the SWI derived from SMAP SSM using the classic characteristic time lengths (15 days for the 0–25 cm layer and 20 days for the 0–100 cm layer). The simulated GW storage showed a slightly delayed variation relative to the derived SWI. The quantified optimal characteristic time length $T_{opt}$ for SWI estimation (by matching the variations in SMAP-derived SWI and modeled root zone SM) is comparable to $T_{opt}$ obtained in other agricultural regions around the world. This work demonstrates SMAP SM measurements as a potentially useful aid when predicting root zone SM and GW dynamics and validating fully integrated hydrologic models across different spatial scales. This study also provides insights into the dynamics of near surface–subsurface water interaction and the capabilities and approaches of satellite-based SM monitoring and high resolution fully-integrated hydrologic modelling.

## 1 Introduction

Accurate information on soil moisture (SM) and groundwater (GW) storage is essential for assessing water resources and making informed decisions for effective water resource management. SM can be monitored and measured using ground-based in situ sensor networks and remote sensing methods (e.g., Dobriyal et al., 2012). The in situ SM monitoring networks are able to provide continuous measurements for different soil depths or profiles; however, the monitoring sites are typically sparse, especially at continental or global scales, causing difficulty in large-scale spatially distributed SM estimation (e.g., Jonard et al., 2018; Singh et al., 2019). Advances in satellite Earth observation have opened up opportunities for the large-scale and global monitoring of SM at fine to medium resolution (e.g., Bartalis et al., 2007; Entekhabi et al., 2010; Kerr et al., 2010; Njoku et al., 2003; Owe et al., 2008; Xu et al., 2014), but satellite remote sensing only measures the near-surface soil layer (the topmost few centimeters) and cannot directly observe the deeper soils. Further, NASA's Gravity Recovery and Climate Experiment (GRACE) and GRACE-Follow On (GRACE-FO) have made it possible to track changes in terrestrial water storage (TWS) by detecting Earth's gravitational changes (Tapley et al., 2004). The TWS observations, in combination with model outputs or reanalysis products, can be used to quantify GW storage dynamics (Famiglietti et al., 2011; Rodell et al., 2007, 2009, 2018; Syed, et al., 2008; Thomas and Famiglietti, 2019; Zhu et al., 2022). However, the coarse-scale (a native resolution of ~3 degrees in both latitude and longitude) monthly TWS changes provided by the GRACE/GRACE-FO observations cannot fully meet the needs for monitoring the variations in GW across different temporal and spatial scales.

As such, the potential of satellite near-surface soil moisture (SSM) measurements for estimating or predicting the variations in root zone SM and GW has received considerable attention over the past decades (e.g., Bouaziz et al., 2020; Ceballos et al., 2005; Ford et al., 2014; Nayak et al., 2021; Paulik et al., 2014; Sutanudjaja et al., 2013; Tian et al., 2020; Wagner et al., 1999; Zhao et al., 2008). One of the key steps for this important application is identification of the coupling strength and the associated temporal differences in response to wetting/drying processes among different subsurface layers, which can vary remarkably across different regions and different time windows depending on a suite of factors, such as depths considered, soil hydraulic properties, soil texture, climate conditions, and land cover (e.g., Albergel et al., 2008; Bouaziz et al., 2020; Wang et al., 2017).

The differences in responses to wetting/drying processes in the soil profile can be examined using in situ measurements (e.g., Mahmood et al., 2012; Wu et al., 2002) or hydrological models (e.g., Mahmood and Hubbard, 2007). The time lagged cross-correlation in SM variations identified between the surface and deeper soil layers (e.g., Mahmood and Hubbard, 2007; Mahmood et al., 2012; Wu et al., 2002) may indicate that the deeper subsurface SM variability could be approximated by delaying the temporal variations in SSM. On the other hand, the deeper subsurface soil water content can be estimated by smoothing the SSM time series since soil water in the deeper layers typically exhibit smaller variations and longer response times to critical precipitation/drying events that occur at the surface (e.g., Albergel et al., 2008; Manfreda et al., 2014;

Ragab, 1995; Wagner et al., 1999). A widely used smoothing method is the Soil Water Index (SWI) that estimates the subsurface SM profiles from the SSM time series using an exponential filter with the characteristic time length T as the only control parameter (Wagner et al., 1999). The optimal characteristic time length ($T_{opt}$) can be obtained by matching the SWI to reference root zone SM (e.g., Bouaziz et al., 2020; Ceballos et al., 2005; Ford et al., 2014; Paulik et al., 2014; Tian et al., 2020; Wagner et al., 1999).

Over the past decades, land surface and hydrological models have played an important role in quantifying $T_{opt}$ for SWI estimation. Albergel et al. (2008) investigated the effects of various factors on $T_{opt}$ for SWI estimation by a combined use of in situ and model data for soils in France. Wang et al. (2017) demonstrated the capability of vadose zone model simulations in quantifying the relationships between $T_{opt}$ and its various influencing factors (precipitation, land cover, and soil hydraulic properties) over the continental United States. Bouaziz et al. (2020) utilized the root zone SM simulated by a process-based lumped hydrological model as reference to quantify $T_{opt}$ values for SWI estimation from different satellite SM products across a number of watersheds in France. In addition, the inter-comparisons between satellite SM and modeled SM data have received intensive research efforts (e.g., Al-Yaari et al., 2014; Dorigo et al., 2010; Draper et al. 2013; Parrens et al., 2012).

Recent advances in high resolution fully-integrated surface water-groundwater modeling for Canadian basins (Erler et al., 2019; Frey et al., 2021; Xu et al., 2021; Aziz et al., 2023) have provided new opportunities for simulating water dynamics in the variably-saturated subsurface domain. Such models present better ability to reproduce realistic root zone SM and GW dynamics than surface-water models used in previous studies. Hence, these models are well suited to help expand our understanding of connections between satellite SM and the variably-saturated subsurface flow regime. Accordingly, this study aims to advance our understanding of: i) the coupling and response time differences between satellite SM dynamics and transient soil water and groundwater storage characteristics, ii) the dependence of coupling and response time differences on spatial scale, iii) the ability of state-of-the-art satellite SM monitoring to predict root zone SM and GW dynamics, and iv) the ability of satellite SM data to assist with validation of large-scale integrated hydrologic models. To this end, the study herein examines the linkages between the Soil Moisture Active Passive (SMAP) SM, which represents one of the state-of-the-art satellite-based SM products, and the subsurface SM and GW dynamics simulated by a high-resolution fully-integrated surface water-groundwater model of an agriculture-dominated watershed across two spatial scales, namely SMAP 9-km grid cell and watershed.

## 2 Data and Methods

### 2.1 Study Watershed

The study domain is the South Nation Watershed (SNW) in eastern Ontario, Canada (Fig. 1a). The SNW is an agriculture-dominated, mixed use watershed with an areal coverage of about 3900 km$^2$ (Figure 1b).

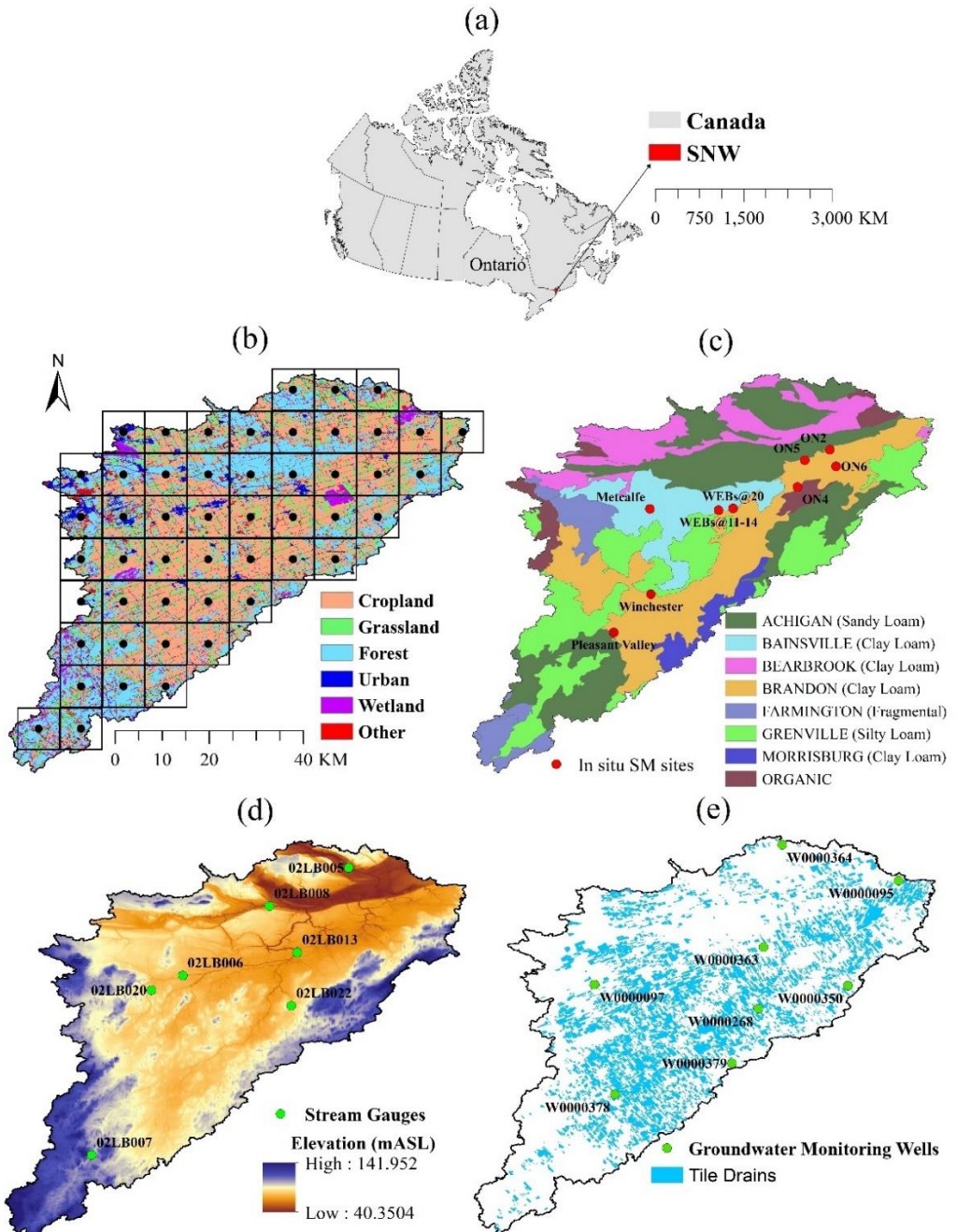

**Figure 1:** (a) Location of the South Nation Watershed (SNW). (b) The SNW land cover map (source: Agriculture and Agri-Food Canada's Annual Crop Inventory 2015) overlaid with location of the 9-km grids (boxes for the pixels and black dots for the centers) for the SMAP SM product used in this study. (c) Soil map for the SNW (source: Soil Landscapes of Canada version 3.2, Agriculture and Agri-Food Canada), along with the in situ SM monitoring sites. (d) Surface elevation along with location of streamflow gauges. (e) Tile drains installed for the SNW (provided by the South Nation Conservation Authority), along with location of groundwater level monitoring wells.

The major crop types grown in the area are corn and soybeans. In the agricultural fields, natural soil water drainage is
typically slow on account of extensive clay loam soil (Fig. 1c). The agricultural region generally has low topographic relief
(Fig. 1d), with artificial subsurface drainage (tile drains) to drain excess water from fields to facilitate crop productivity (Fig.
1e). The tile drains tend to be spaced about 15 to 17 meters apart (Sunohara, et al., 2015), with a total tile-drained area of 956
$km^2$ (about 25% of the watershed). The watershed is characterized as having a humid temperate climate, and twenty-year
(1998–2017) average annual precipitation is about 1000 mm, and average annual evapotranspiration is about 600 mm. The
average water table is 1 to 3 meters below surface across much of the agricultural landscape.

## 2.2 Satellite SSM and SWI

For this study, satellite-based SM retrievals are taken from the SMAP enhanced L3 radiometer 9 km EASE-grid SM
(SPL3SMP_E) version 5 product (O'Neill et al., 2021). The SPL3SMP_E product provides daily composite estimates of
near-surface (~ top 5 cm) SM at a resolution of 9 km, retrieved from the AM (descending half orbits) and PM (ascending
half orbits) brightness temperatures observed by the SMAP radiometer. In this study, only the SPL3SMP_E AM retrievals
are used since the AM product is superior to the PM product over the study region, which is consistent with the AM/PM
product comparison over the Great Lakes region (Xu, 2020). The location of the SPL3SMP_E product 9-km grid cells over
the SNW is illustrated in Fig. 1b. A filtering step (using the SMAP product's ancillary information) was conducted to
remove the SMAP SM estimates that were affected by various adverse factors (e.g., open water, frozen surfaces, snow, rain,
or radio frequency interference). This study uses the SMAP SM data between March 31, 2015 (the date when SMAP started
operation) and December 31, 2017, which is the time span overlapping with the temporal coverage of model simulations
used in this work (section 2.3).

The SMAP SSM can be used to derive the moisture content in the deeper layers or entire unsaturated root zone with the SWI
approach, which estimates the subsurface SM as a function of SSM utilizing an exponential filter (Wagner et al., 1999). The
SWI method considers a near-surface soil layer and a subsurface layer. A water-balance approach is then applied to the two
soil layers to compute the water fluxes across them, which are assumed to be proportional to their SM differences. In this
study, we use the recursive exponential filter (Albergel et al., 2008), which is suitable for SWI estimation from the SSM
observed at irregular time intervals. The SWI at time $t_i$ is given by Eq. (1):

$$\text{SWI}(t_i) = \text{SWI}(t_{i-1}) + K(t_i)\left(\text{SSM}(t_i) - \text{SWI}(t_{i-1})\right), \qquad (1)$$

where $\text{SWI}(t_i)$ and $\text{SWI}(t_{i-1})$ denote the SWI values at time $t_i$ and $t_{i-1}$, respectively; $\text{SSM}(t_i)$ represents the SMAP near-surface soil moisture at time $t_i$; and $K(t_i)$ is the gain at time $t_i$, which is given in a recursive form as in Eq. (2):

$$K(t_i) = \frac{K(t_{i-1})}{K(t_{i-1}) + exp\left(\frac{-(t_i - t_{i-1})}{T}\right)}, \qquad (2)$$

where $K(t_{i-1})$ is the gain at time $t_{i-1}$. The gain $K$ ranges from 0 to 1 with the initialization $K(t_0) = 1$, while SWI is initialized using the SSM series, i.e., $\text{SWI}(t_0) = \text{SSM}(t_0)$. The parameter $T$ is the characteristic time length in days, and can be considered as a surrogate for many factors (e.g., soil depth, soil properties, evaporation, and runoff) that can influence SM changes due to drying and wetting processes (Albergel et al., 2008; Wagner et al., 1999).

## 2.3 Fully-Integrated Surface Water-Groundwater Model

The high-resolution fully integrated surface water–groundwater simulations are conducted using HydroGeoSphere (HGS) (Aquanty, 2022, Hwang et al., 2014; Frey et al., 2021). HGS uses the one-dimensional (1D) Manning's open channel flow equation to govern river/stream flow, the diffusion wave equation to govern two-dimensional (2D) overland flow, and the Richards' equation to govern three-dimensional (3D) variably-saturated subsurface flow. The channel/surface/subsurface regimes naturally interact with each other through the exchange of water fluxes in response to varying pressure gradients. Unlike loosely or sequentially coupled groundwater-surface water models, HGS is a fully-integrated model, providing the simultaneous solution of the channel, surface and subsurface flow regimes at each time step. Detailed information on HGS can be found in the relevant documents (Aquanty, 2022, Hwang et al., 2014; Frey et al., 2021).

Within the model there are seven subsurface layers that are composed of 3D triangular prisms, formed by superimposing eight mesh layers of planar elements from the soil surface downward to a depth of ~35 m. In total, there are 171,609 planar elements per mesh layer, equating to a total 1,201,263 3D elements across the seven-layer subsurface domain. The 3D unstructured finite element mesh that underpins the HGS model carries 125 m spatial resolution along Strahler 2+ streams and rivers, and up to 375 m resolution in areas distal to the resolved surface water features. The 2D overland flow domain (composed of planar elements) and the 1D channel domain (composed of linear elements) are both superimposed onto the subsurface 3D domain. In the model subsurface domain, the three soil layers (0–25 cm, 25–50 cm, and 50–100 cm depths) of 3D elements were constructed by superimposing the four mesh layers of planar elements at the soil surface, 25 cm, 50 cm, and 100 cm depths, respectively. The four mesh soil layers of planar elements can provide the simulated SM at the four specific depths (soil surface, 25 cm, 50 cm, and 100 cm), while the three soil layers of 3D elements can provide the simulated SM for the depth intervals of 0–25 cm, 25–50 cm, and 50–100 cm. The simulated SM from the planar element mesh layer at the soil surface represents the simulated SSM. Underlying the three soil layers are the four hydrostratigraphic layers (three Quaternary layers and one bedrock layer), with geometry and lithology derived from Logan et al. (2009). The tile drains installed in the SNW (Fig. 1e) are not resolved in the model because of resolution constraints associated with the size of the model domain and the necessity of carrying a practical number of finite elements. The influence of tile drainage absence in the model will be discussed in this study (section 6.3).

Appropriate spin-up is essential for integrated surface-subsurface models (e.g., Ajami et al., 2014, 2015; Erdal et al., 2019). Similar to Frey et al. (2021), the HGS model herein was initialized following a three-step procedure. Firstly, the model was

forced by long-term (~30-year) average annual net precipitation until steady-state groundwater heads and streamflow rates were established. Secondly, using steady-state as an initial condition, the model was forced by monthly normal liquid water flux and potential evapotranspiration for a decadal cycle, yielding a year over year dynamic equilibrium condition. Thirdly,

using dynamic equilibrium as an initial condition, the model was forced with daily transient liquid water flux and potential evapotranspiration derived from Natural Resources Canada (NRCan)'s gridded daily climate data sets (McKenney et al., 2011) in combination with snow water equivalent data derived from the ERA5 land surface reanalysis product (Muñoz-Sabater et al., 2021). The daily transient simulations extended from January 1, 2008 to December 31, 2017, and were run multiple times, with model performance only evaluated after the second set of simulations. The model calibration primarily

involved manually tuning the soil hydraulic conductivity and the Manning's surface roughness coefficient for the 1D river/stream channels. The objective of calibration was to optimize surface water flow rates at the hydrometric stations (Fig. 1d) and groundwater levels at the monitoring wells (Fig. 1e). Subsequent analysis is based on the March 31, 2015 (the date when SMAP started operation) to December 31, 2017 time frame, using daily transient output data from the calibrated HGS model.

**2.4 Comparison Analysis and Performance Metrics**

**2.4.1. Evaluation of SMAP and Modeled Soil Moisture**

The SM estimates from the SMAP product and HGS model simulations are evaluated against the SM measurements from in situ monitoring sites (Fig. 1c). The specification of in situ SM measuring is provided in Table 1. Since the in situ SM sites are sparse, the evaluation is available only at point-scale. SMAP SSM and HGS simulated SSM estimates are evaluated

using the 0–5 cm in situ SM measurements. The HGS simulated root zone SM is evaluated at two depth profiles: 0–25 cm and 0–100 cm. In the 0–25 cm layer, the simulated SM in the model's top soil layer (0–25 cm) is compared to a depth-weighted average of in situ measurements in the top 25 cm soil (i.e., 5 cm and 25 cm depths at the RISMA sites, 10 and 20 cm depths at Metcalfe and Pleasant Valley, and 20 cm depth at Winchester stations, see Table 1). In the 0–100 cm layer, a depth-weighted average of simulated SM from the model's three soil layers (0–25 cm, 25–50 cm, and 50–100 cm depths) is

evaluated against a depth-weighted average of in situ measurements in the top 100 cm soil (i.e., 5, 20, and 50 cm depths at the RISMA sites; 10, 20, and 50 cm depths at Pleasant Valley, and 15 and 45 cm depths at WEBS stations, see Table 1). At each in situ site, the unbiased Root Mean Squared Error (ubRMSE) and Pearson correlation coefficient ($R$) are computed based upon the daily time series using the following equations:

$$ubRMSE = \sqrt{E[((\theta_s - E[\theta_s]) - (\theta_i - E[\theta_i]))^2]} \ , \qquad (3)$$

$$R = E[(\theta_s - E[\theta_s])(\theta_i - E[\theta_i])](\sigma_s \sigma_i)^{-1} \qquad , \qquad (4)$$

where $E\,[\bullet]$ is the expectation operator. $\theta_s$ and $\theta_i$ indicate the daily time sequences of satellite (or model) soil moisture and in situ data, respectively. $\sigma_s$ and $\sigma_i$ denote the standard deviations of $\theta_s$ and $\theta_i$, respectively.

**Table 1**. Specification of in situ soil moisture stations

| Station ID | Latitude (°) | Longitude (°) | Sampling Intervals | Measuring Depths (cm) | Data Period |
|---|---|---|---|---|---|
| [a]RISMA ON2 | 45.4016 | -74.9479 | 15 mins | 0-5, 5, 20, 50 | 2015 to 2017 |
| RISMA ON4 | 45.3140 | -75.0193 | 15 mins | 0-5, 5, 20, 50 | 2015 to 2016 |
| RISMA ON5 | 45.3769 | -75.0031 | 15 mins | 0-5, 5, 20, 50, 100 | 2015 to 2017 |
| RISMA ON6 | 45.3628 | -74.9342 | 15 mins | 0-5, 5, 20, 50 | 2016 to 2017 |
| Metcalfe | 45.2626 | -75.3439 | Hourly | 10, 20 | 2015 to 2017 |
| Pleasant Valley | 44.9726 | -75.4237 | Hourly | 10, 20, 50 | 2017 |
| [b]WEBs@11-14 | 45.2598 | -75.1929 | 15 mins | 15, 45 | 2015 to 2017 |
| WEBs@20 | 45.2639 | -75.1607 | 15 mins | 15, 45 | 2015 to 2017 |
| Winchester | 45.0623 | -75.3418 | Hourly | 20 | 2017 |

a. Agriculture and Agri-Food Canada's Ontario Real-Time In-Situ Soil Monitoring for Agriculture (RISMA) stations
b. Agriculture and Agri-Food Canada's WEBs meteorological stations

### 2.4.2 Evaluation of HGS Simulated Streamflow and GW Level

The simulated streamflow rates and GW levels in the fully-integrated modeling framework are physically linked to SM and GW flow, and are hence also a reflection of water dynamics in the variably-saturated subsurface domain. The simulated streamflow is evaluated using streamflow measurements from Water Survey of Canada (WSC) hydrometric stream gauges (Fig. 1d), and performance is assessed with the Nash-Sutcliffe efficiency (NSE) in Eq. (5):

$$NSE = 1 - E[(Q_{obs} - Q_{sim})^2]/E[(Q_{obs} - E[Q_{obs}])^2] , \qquad (5)$$

where $E\,[\bullet]$ is the expectation operator. $Q_{obs}$ and $Q_{sim}$ indicate the daily time sequences of observed and simulated stream discharge values, respectively. NSE ranges from $-\infty$ to 1 with 1 as the optimal value.

The simulated GW levels are compared to GW level measurements provided by the Provincial Groundwater Monitoring Network (PGMN, https://data.ontario.ca/dataset/provincial-groundwater-monitoring-network) wells (Fig. 1e). Since the temporal variability information is of the most interest for the simulated GW levels in this study, the Pearson correlation coefficient ($R$) between the temporal variations of simulated and observed GW level anomalies is calculated at each GW monitoring well across the SNW. The GW level anomalies represent the departures from their respective average over the evaluation period (March 31, 2015 to December 31, 2017).

### 2.4.3  Comparison between SMAP and HGS Model Simulations

The SMAP data (SSM and SWI) are compared to the HGS model simulations (SSM, subsurface SM, and GW storage) to quantify the vertical coupling and response time differences between satellite SM and the variably-saturated subsurface

water. The comparisons are made at both the 9-km (SMAP product grid) resolution and the entire watershed, and are measured using the unbiased Root Mean Squared Difference (ubRMSD), $R$, anomaly $R$, and Spearman's rank correlation ($\rho$), depending upon the variables under comparison. The ubRMSD and $R$ are computed using the equations similar to Eq.
(3) and Eq. (4), but with the two variables from the SMAP and HGS simulation, respectively. The anomaly $R$ calculation is similar to the $R$ calculation, but uses the anomaly time series of the variables, which are defined as departures of raw values from their monthly normals over the study period (2015–2017). For each variable, all three-year (2015 to 2017) monthly data must be valid for computing the monthly normal of a calendar month. The Spearman's rank correlation ($\rho$) is calculated as,

$$\rho = 1 - \frac{6 \sum d^2}{n(n^2-1)}, \tag{6}$$

where $d$ represents the difference between the ranks of the SMAP and HGS model variables, and $n$ denotes the length of data. In this study, $\rho$ is used for time scale quantification for water transport from the surface soil layer to deeper unsaturated/saturated zones.

## 3 Evaluation of SMAP SSM and Model Simulations

The evaluation scores for the SMAP SSM and HGS SM across the individual in situ sites are listed in Table 2. In this study,
the in situ SSM (0–5 cm) measurements are only available at Agriculture and Agri-Food Canada's Ontario Real-Time In-Situ Soil Monitoring for Agriculture (RISMA) stations (Tabel A1). Fig. 2 shows the SSM time series from SMAP, HGS model, and in situ measurements at the four Ontario RISMA stations. Overall, both the SMAP and HGS modeling captured the in situ observed SSM temporal variability very well (Fig. 2). Both the SMAP SSM and simulated SSM showed a mean ubRMSE of about 0.05–0.06 $\mathrm{m^3\,m^{-3}}$ and a mean $R$ of about 0.7 with the in situ measurements (Table 2). The performance of
SMAP SSM over the SNW is very similar to that over the Great Lakes region (Xu, 2020; Xu and Frey, 2021), which is approximately adjacent to the study region SNW.

**Table 2**  Soil moisture (SM) performance metrics

| Metrics | Station ID | SMAP SSM | HGS SM | | |
|---|---|---|---|---|---|
| | | | Near Surface | 0–25 cm | 0–100 cm |
| ubRMSE ($\mathrm{m^3\,m^{-3}}$) | RISMA ON2 | 0.056 | 0.050 | 0.038 | 0.021 |
| | RISMA ON4 | 0.047 | 0.061 | 0.047 | 0.028 |
| | RISMA ON5 | 0.055 | 0.041 | 0.042 | 0.022 |
| | RISMA ON6 | 0.079 | 0.066 | 0.040 | 0.030 |
| | Metcalfe | — | — | 0.053 | — |
| | Pleasant Valley | — | — | 0.024 | 0.013 |
| | WEBs@11-14 | — | — | — | 0.023 |

|   | | | | | |
|---|---|---|---|---|---|
| | WEBs@20 | — | — | — | 0.035 |
| | Winchester | — | — | 0.033 | — |
| | **Average** | **0.059** | **0.054** | **0.040** | **0.025** |
| | | | | | |
| | RISMA ON2 | 0.67 | 0.76 | 0.74 | 0.76 |
| | RISMA ON4 | 0.72 | 0.56 | 0.67 | 0.73 |
| | RISMA ON5 | 0.66 | 0.82 | 0.75 | 0.86 |
| | RISMA ON6 | 0.71 | 0.80 | 0.82 | 0.76 |
| *R* | Metcalfe | — | — | 0.70 | — |
| | Pleasant Valley | — | — | 0.81 | 0.77 |
| | WEBs@11-14 | — | — | — | 0.88 |
| | WEBs@20 | — | — | — | 0.70 |
| | Winchester | — | — | 0.77 | — |
| | **Average** | **0.69** | **0.74** | **0.75** | **0.78** |

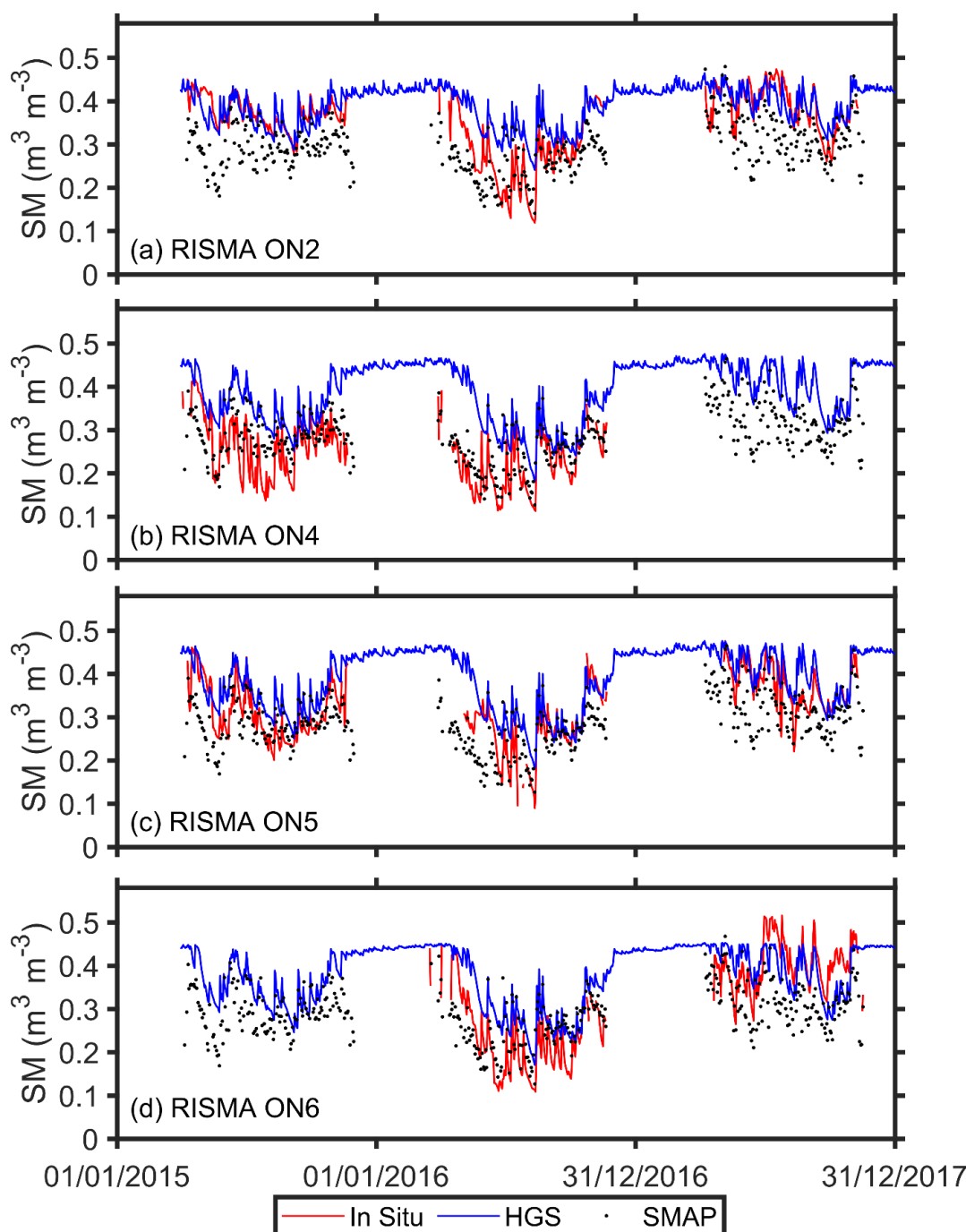

**Figure 2:** The SSM time series from SMAP, HGS model simulation, and in situ measurements, respectively, at the four RISMA stations: (a) ON2, (b) ON4, (c) ON5, and (d) ON6.

Figs. A1 and A2 show the root zone SM time series comparison between HGS simulations and in situ measurements for the
0-25 cm and 0-100 cm soil layers, respectively. Overall, the HGS simulations agree with the in situ measurements very well
in terms of the SM temporal variability in the two soil profiles across all available validation sites (Figs. A1 and A2), with
the mean $R$ close to or exceeding 0.75 for both soil profiles (Table 2). Unsurprisingly, the ubRMSE for the HGS soil
moisture decreases with the increasing soil depth (Table 2), resulting from a smaller soil moisture temporal variability in a
deeper profile (e.g., Albergel et al., 2008; Xu, 2020).


Fig. A3 presents the simulated and observed hydrographs along with the calculated NSE values at the seven WSC
streamflow gauges across the study watershed (as shown in Fig. 1d). The HGS simulations performed well in capturing the
timing of peak flows. The NSE values are typically high, exceeding 0.62 for all gauges, although the underestimation of
peak flows is also evident in the HGS hydrographs. A possible explanation for the flow underestimation is that the tile drain
flow, which was not resolved in the present HGS model simulations, is also a source of discharge for streams (due to a
shallow water table) in the real-world SNW. Further, the comparisons between the simulated and observed GW level
anomalies at the eight GW monitoring wells were provided in Fig. A4. In general, the GW temporal variability was well
reproduced by the HGS modelling across the monitoring wells, with $R$ ranging from 0.4 to 0.86.

Overall, the SMAP SM product can capture the SSM variability well over the study region, while the HGS simulations
match the observed surface/subsurface water dynamics well. This supports the HGS model's application towards quantifying
the dynamic behavior of surface/subsurface hydrologic conditions, and to testing linkages between SMAP measurements and
simulated water content in the variably saturated subsurface.

## 4. Comparisons between SMAP SSM and HGS Model Simulations

### 4.1 Comparison at the 9-km Scale

In this section, we compare the SMAP SSM with the HGS simulated SM at the SMAP product grid (9-km) scale. Since the
HGS model has a higher resolution than the 9-km SMAP grid (see section 2.3), the model SM estimates are spatially
aggregated (i.e., averaged) within each SMAP grid cell. Fig. 3a illustrates the ubRMSD values across all SMAP grids for the
SMAP SSM and HGS SSM comparison, with the summarized ubRMSD provided in Fig. 4a. The SMAP grid-scale
ubRMSD values range from 0.04 to 0.06 $m^3 m^{-3}$ and are typically lower in the forested areas than over the agricultural fields
(Fig. 3a with the land cover map provided in Fig. 1b). The average ubRMSD between the SMAP and HGS SSM estimates is
about 0.047 $m^3 m^{-3}$ at the 9-km scale (Fig. 4a).

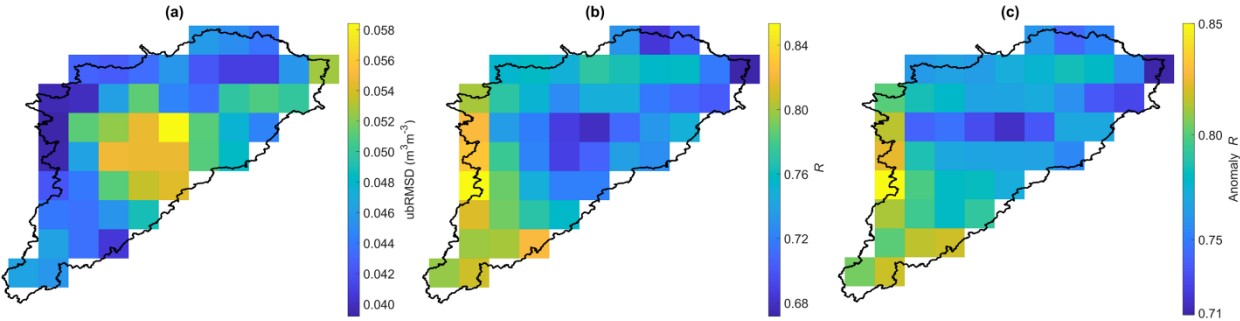

**Figure 3:** (a) ubRMSD, (b) *R*, and (c) anomaly *R* between the SMAP SSM and HGS SSM across all SMAP grids.

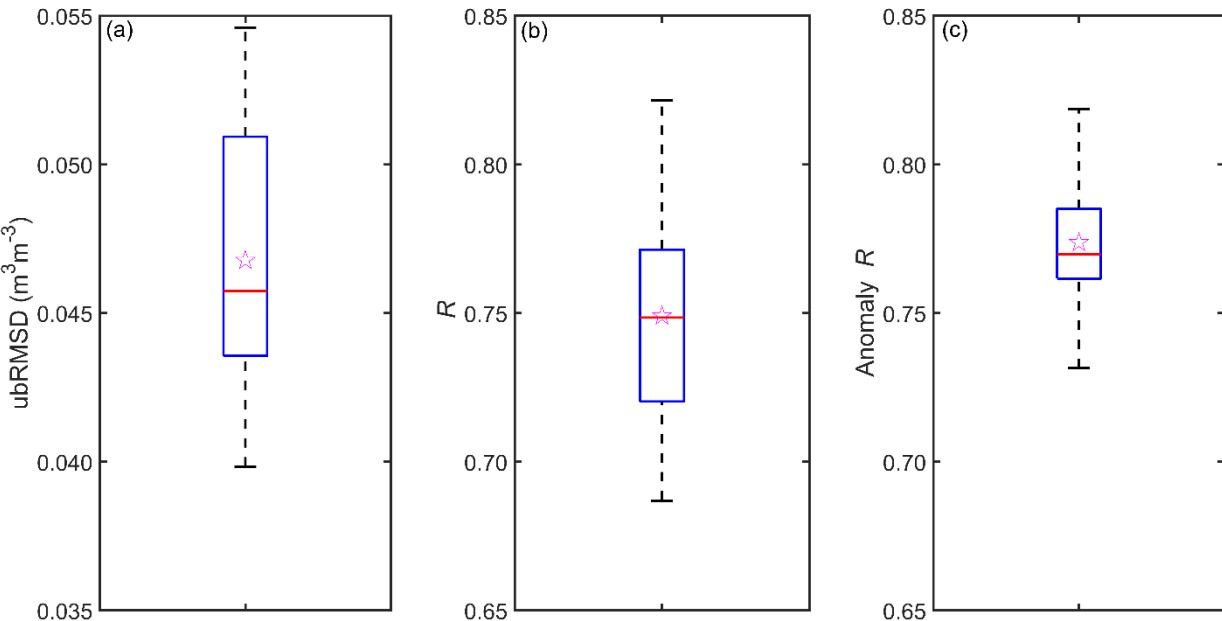

**Figure 4:** Boxplots of (a) ubRMSD, (b) *R*, and (c) anomaly *R* between the SMAP SSM and HGS SSM, summarized from all 9-km grids within the study watershed. Within each box, the horizontal segment and the star denote the median and mean of the sample data, respectively; the lower and upper edges of the box indicate the 25th and 75th percentiles, respectively; and the bottom and top ends of the whiskers denote the 5th and 95th percentiles, respectively.

Fig. 3b shows the *R* values between the SMAP SSM and the HGS simulated SSM across all individual SMAP grids, which are summarized using the boxplot in Fig. 4b. The simulated SSM can capture the SMAP observed SSM dynamics quite well

at the 9-km grid scale with an average *R* exceeding 0.7 (Fig. 4b). In terms of the spatial variability, the *R* values are typically higher over the forests than their counterparts over the agricultural lands (Fig. 3b), which may be in part due to tile drainage not fully captured by the model (see section 6.3). The anomaly *R* results (Figs. 3c and 4c) are similar to the *R* results, indicating that the obtained correlations between SMAP and the HGS model are dominated by the day-to-day variations

(rather than the seasonal trends) in the SM time series. Further, the linear regression between SMAP SSM (independent

variable) and HGS SSM (dependent variable) suggests that modeled SSM is systematically wetter than SMAP SSM (intercept > 0) but shows a smaller change in response to every unit change in SMAP SSM (slope < 1) across the watershed (Figs. A5 and A6).

To examine the linkage between SMAP SSM and water storage variability in the deeper subsurface, time lagged cross-
correlations between the SMAP SSM and simulated subsurface SM and GW storage were calculated for each 9-km grid. Here the Spearman's rank correlation (rather than the Pearson correlation) is used for the time lagged cross-correlation analysis since the monotonic (rather than linear) relationship is of the most interest for identifying the phase difference between near-surface and deeper subsurface water content variability. Fig. 5 presents the Spearman's rank correlations (the 5th to 95th percentiles from all SMAP grids over the study watershed) between the time series of SMAP SSM and the HGS
simulated subsurface SM (0–25 cm, 25–50 cm, and 50–100 cm depths) and GW storage for time lag ranging from 0 to 60 days. The optimal time lags (in days) and corresponding highest Spearman's rank correlations across all SMAP grids are provided in Figs. A7 and A8, respectively.

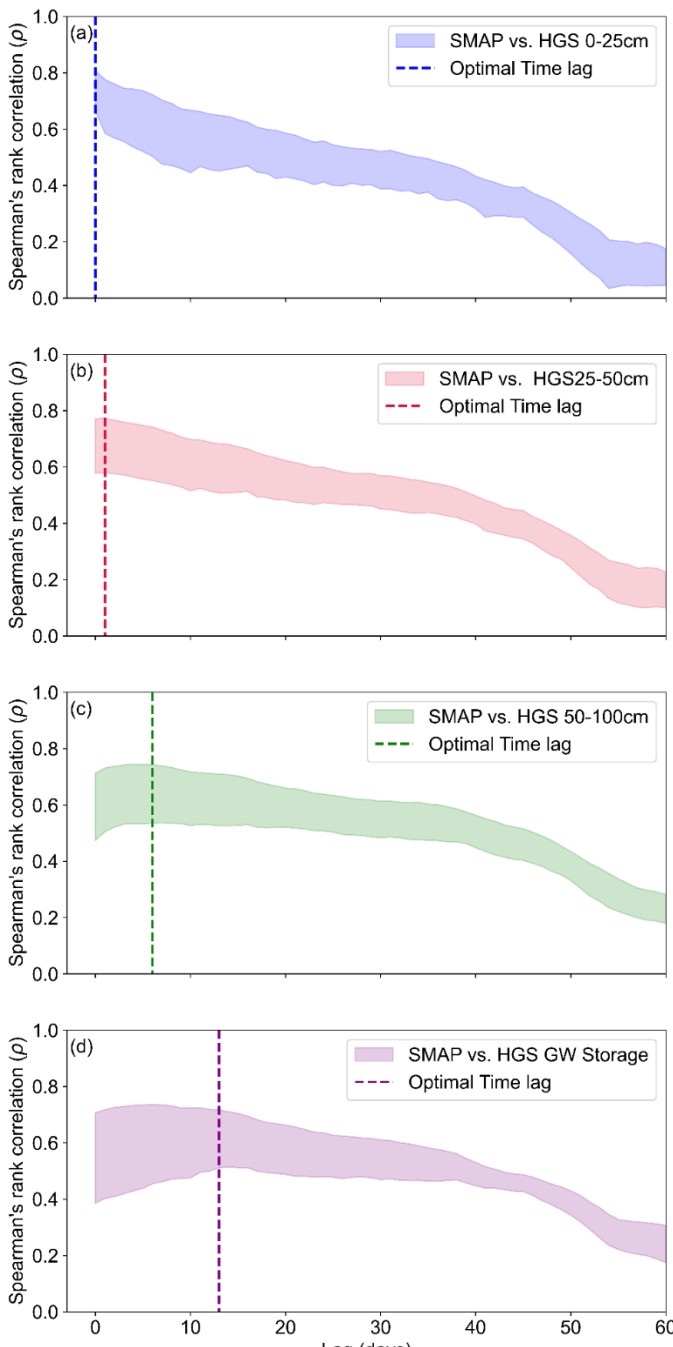

Figure 5: Spearman's rank correlation coefficients between the time series of SMAP SSM and HGS simulated SM from (a) 0–25 cm depth, (b) 25–50 cm depth, and (c) 50–100 cm depth, respectively, as well as (d) GW storage for time lags ranging from 0 to 60 days. Positive lags indicate that the SMAP data are leading the HGS simulations. In each plot, the shaded band represents the 5th to 95th percentiles of the results from all SMAP grids within the study watershed. For each SMAP grid, the optimal time lag is defined as the one with the maximum Spearman's rank correlation between SMAP SSM and HGS simulated variable. The vertical dashed line indicates the median of the optimal time lags from all SMAP grids.

Unsurprisingly, the simulated SM in the 0–25 cm depth (Figs. 5a and A7a) showed simultaneous response (a time lag of 0 day) to the SSM variability captured by SMAP across the study watershed. By contrast, simulated water content variations in the deeper subsurface showed a delayed response relative to the SMAP SSM variability. The optimal time lag increased with depth, with a median delay of about 1–2 days for the 25–50 cm SM (Fig. 5b), about 6 days for the 50–100 cm SM (Fig. 5c), and about 11–12 days for the GW storage (Fig. 5d).

By comparing the spatial distribution of time lags on a SMAP grid cell basis (Fig. A7) to the soil distribution (Fig. 1c), the time delay for deeper zones is typically shorter in regions with well drained soils (e.g., the southwestern portion of SNW) than in areas with poorly or imperfectly drained soils (e.g., the northern SNW), reasonably reflecting the impact of soil properties on deeper subsurface hydrologic response. Table A1 provides the average optimal time lags for the six major soils over the study watershed. For each soil, the averaged optimal time lag is calculated using the 9-km SMAP grids dominated by the soil texture (the Organic and Morrisburg soils are not calculated and included in the table due to their insufficient sample grids). Clearly, the soil drainage has a key impact on the spatial variability of the time lags for deeper layers. The optimal time lag for the 25–50 cm depth is statistically shorter (longer) than 1 day in regions with well drained (imperfectly or poorly drained) soils. Moving to the 50–100 cm depth, on average, the soils of Achigan (imperfectly drained) and Bearbrook (poorly drained) dominated regions experienced the longest optimal time delay (close to or higher than 10 days). Further, the optimal time delay is statistically less (more) than 10 days for the GW system in the areas with good (poor or imperfect) soil drainages. It should also be noted that the quantified time delay in deep subsurface water dynamics did not explicitly account for the impact of tile drainage due to the absence of tile drains in the model. The maximum correlations (corresponding to the optimal time lags) between the SMAP SSM and simulated subsurface water also showed a clear spatial pattern, with higher values in the southwestern SNW (Fig. A8), which corresponds to the regions with well drained soil (Grenville and Farmington soils in Fig. 1c). Overall, the soil texture showed an important impact on the vertical coupling length (correlations) and response time differences between satellite SSM and the variably-saturated subsurface water.

## 4.2 Comparison at the Watershed Scale

Fig. 6 compares the SMAP and HGS simulated time series for the watershed-averaged SSM. Although the simulated SSM is systematically wetter than SMAP SSM, the simulated results match the SMAP measurements very well in terms of SSM variations, with both the $R$ and anomaly $R$ between them exceeding 0.8 and an ubRMSD less than 0.04 $m^3\,m^{-3}$. The observed mean biases between the SMAP and modelled SSM may in part be related to the calibration of the SMAP SM retrieval algorithm. Although the SMAP SM retrievals can capture the SSM variability very well, they typically show an underestimation of SSM (i.e., drier surface soils) over Canadian agricultural regions due to issues with correcting the effects of growing vegetation (e.g., Colliander et al., 2017). In addition, the absence of tile drainage in the HGS model could cause a wet bias over the tile drained landscape (~25% of the entire watershed) and therefore moderately increase the wetness of

the simulated watershed-averaged SSM. Given the scarcity of in situ SM measurements, this study is unable to investigate whether the SMAP retrieval algorithm or the modeling should be blamed for the bias. However, such bias has a negligible effect on this study since the SM temporal variations from the SMAP and HGS model are of primary interest herein.

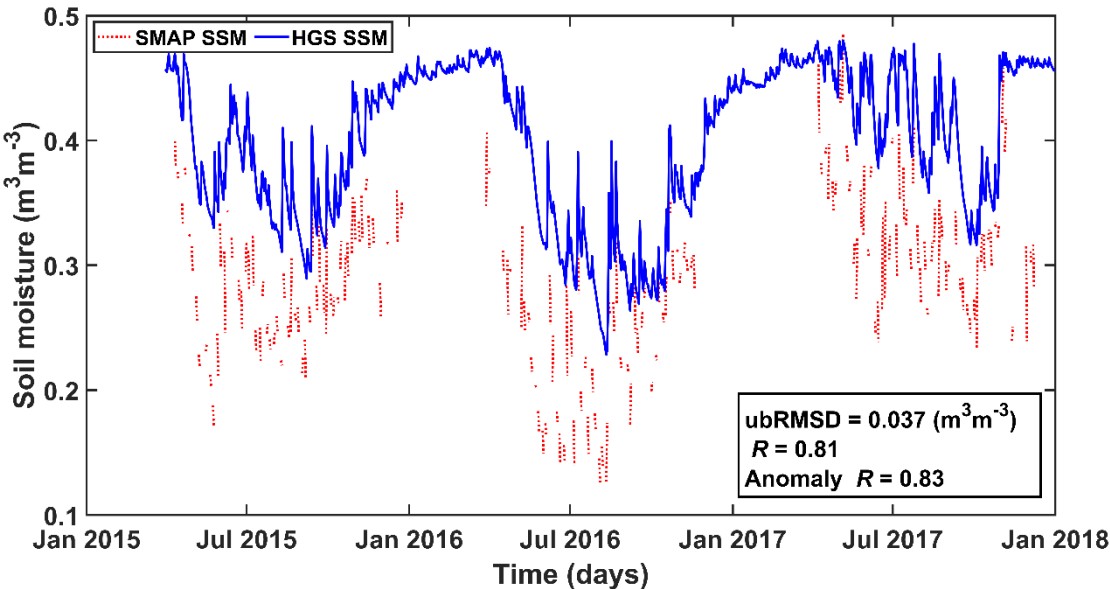

**Figure 6:** Comparison between the SMAP and HGS simulations for the watershed averaged SSM time series.

The relationships between variations of SMAP SSM and HGS simulated water content in deeper unsaturated/saturated zones at the watershed scale are quantified in Fig. 7. Fig. 7a shows the time series of the simulated watershed-averaged SM in the 0–25 cm, 25–50 cm, and 50–100 cm depths and watershed integrated GW storage, in comparison with the watershed-averaged SMAP SSM. The Spearman's rank correlations between the SMAP SSM and the HGS simulated subsurface water for time lags ranging from 0 to 60 days are provided in Fig. 7b. Variations in simulated subsurface water are highly correlated across the different depth intervals. The surface soil layer (0–25 cm) is directly impacted by influxes and effluxes of water and therefore shows the largest day-to-day water content variability, while the 50–100 cm SM and GW storage show comparably smoother day-to-day fluctuations (Fig. 7a). Accordingly, in Fig. 7b, variations in time-lagged correlation between SMAP SSM and simulated subsurface water become smoother as the subsurface depth increases, which also reflects the gradual filtering of high-frequency signals in subsurface water content with the increasing depth.

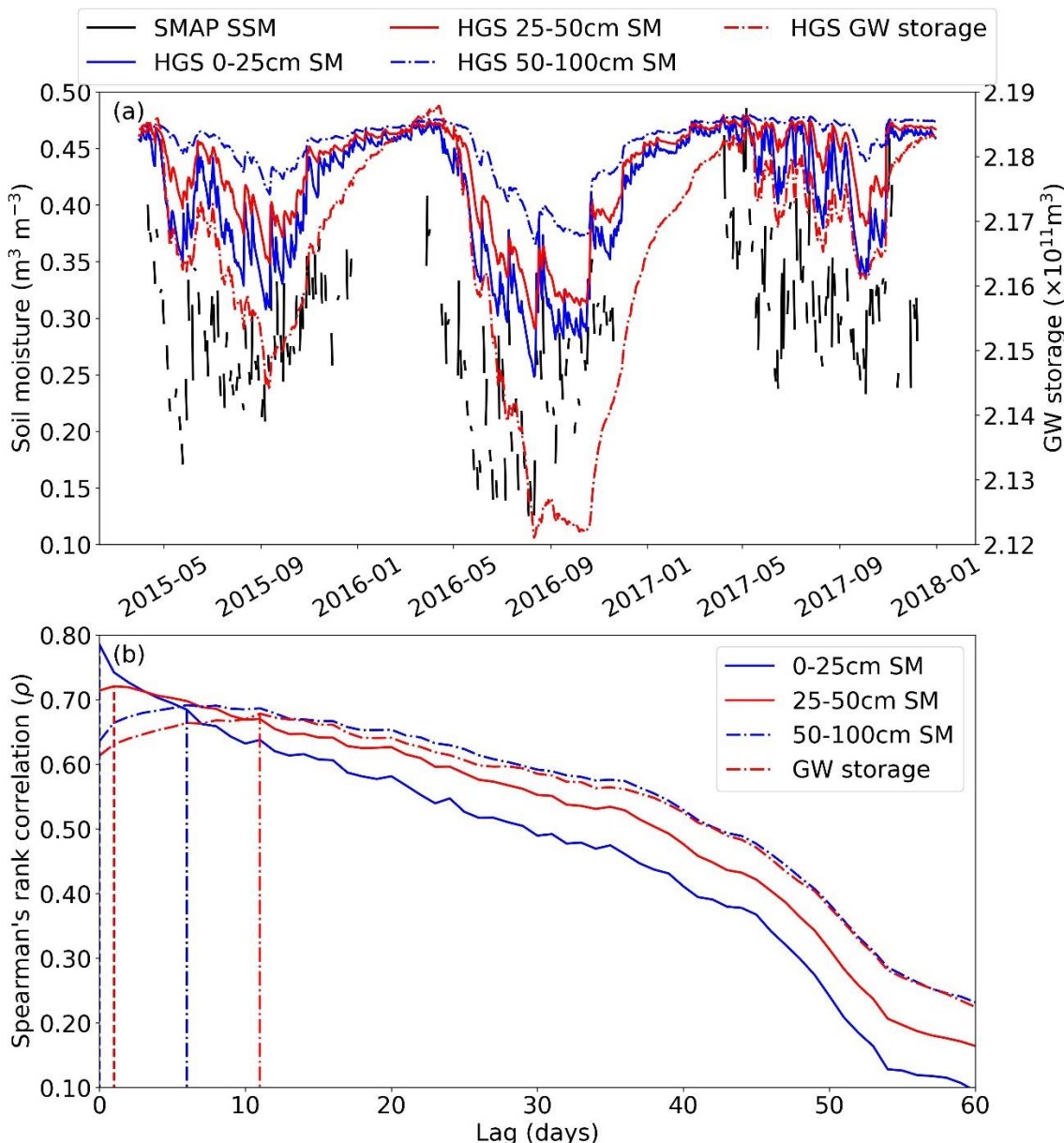

**Figure 7:** (a) Daily time series of HGS simulated watershed-averaged SM in the 0–25 cm, 25–50 cm, and 50–100 cm depths, and watershed integrated GW storage, along with watershed-averaged SMAP SSM. (b) Spearman's rank correlation between the watershed-averaged SMAP SSM versus the HGS simulated watershed-averaged 0–25 cm SM, 25–50 cm SM, 50–100 cm SM, and watershed integrated GW storage, respectively, for time lag ranging from 0 to 60 days. Positive lags indicate that SMAP data lead the HGS simulations. In (b), for each pair of comparisons, a vertical dashed line is provided to indicate the location of the optimal time lag corresponding to the maximum Spearman's rank correlation.


A very good agreement is observed between the variations of the watershed-averaged SMAP SSM and HGS simulated watershed-averaged surface layer (0–25 cm) SM with their highest Spearman's rank correlation coefficient reaching around 0.8, which occurring at a time lag of 0 day (i.e., the delay is less than 1 day and cannot be resolved at daily time steps). With the time lag increasing, the correlation between SMAP SSM and the simulated 0–25 cm SM drops rapidly (green in Fig. 7b). The variations of deeper subsurface SM and GW storage are also in relation to the SMAP SSM variability, but showing a

delayed response. At the watershed scale, the 25–50 cm SM, 50–100 cm SM and GW storage showed the highest Spearman's rank correlation with the SMAP SSM variability at a temporal delay of ~1 day, ~6 days, and ~11 days, respectively (Fig. 7b), which is very similar to the analysis at the 9-km scale (Fig. 5).

In Fig. 7a, the SMAP SSM (top 5 cm) indicated a slightly earlier thaw onset than the model simulated SM in deeper layers.

This reflects a downward heat transfer and migration of thawing front. During a thawing/warming period, the soils typically have a downward temperature gradient (i.e., soil temperature decreases with increased soil depth), which causes a downward heat transfer and migration of thawing front. The thaw onset difference between different depths is consistent with the response time differences between satellite SSM and the subsurface water.

## 5. Comparisons Between SMAP-Derived SWI and HGS Model Simulations

**5.1 SMAP SWI estimation based upon classic time length *T***

Linkage between SMAP SSM-derived SWI and HGS simulations is investigated here. Firstly, SWI based upon the model-independent characteristic time length $T$ (so that the calculated SWI is entirely independent of the model simulations) is compared to simulated subsurface SM. It must be acknowledged that ideally, the time length $T$ (model-independent) should be estimated using in situ SM measurements (e.g., Wagner et al., 1999, Tian et al., 2020). However, given the scarcity of in

situ SM data and the relatively large spatial scale of the analysis herein, it is not possible to determine the time length $T$ (model-independent) based upon evaluation with in situ data. To this end, $T = 15$ days and $T = 20$ days (taken from Wagner et al., 1999), which represent the classic $T$ values for SWI estimation in the surface soil layer (0–20 cm) and the root zone soil layer (0–100 cm), respectively, were used for calculating the model-independent SWI across the 9-km grids. The calculated SWI using $T = 15$ days are compared to the HGS 0–25 cm SM, while the calculated SWI using $T = 20$ days are

compared to the HGS 0–100 cm SM. Fig. A9 presents the ubRMSD, $R$, and anomaly $R$ between SWI and HGS simulated subsurface SM across all 9 km grids. The 9 km grid-scale evaluation metrics are summarized in Fig. 8. Across the SNW, the 9-km scale ubRMSD between SWI and HGS SM typically ranges from 0.03 to 0.05 $m^3\,m^{-3}$ (with an average of 0.035 $m^3\,m^{-3}$) for the top 25 cm layer (Figs. 8a and A9a) and less than 0.04 $m^3\,m^{-3}$ (with an average of about 0.03 $m^3\,m^{-3}$) for the top 100 cm layer (Figs. 8a and A9b). In the two soil depths (top 25 cm and top 100 cm), both the $R$ and anomaly $R$ between SWI and

HGS SM are very high and typically exceed 0.70 (Figs. A9c-f), with their means exceeding 0.82 (Figs. 8b and 8c).

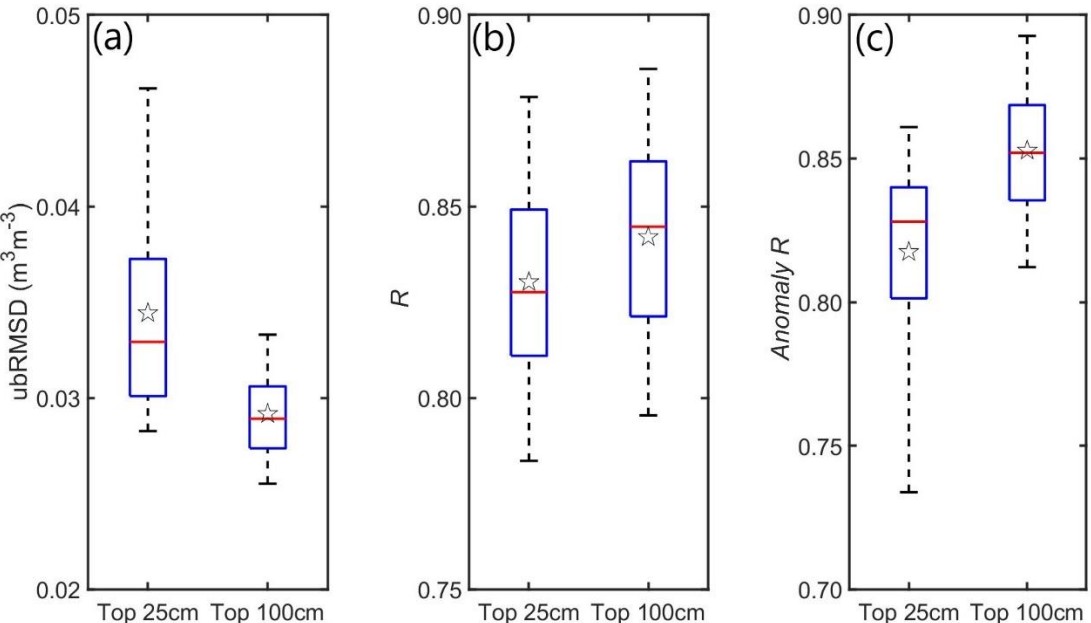

**Figure 8:** (a) Boxplots for the ubRMSD between the SMAP SSM-derived SWI (using $T = 15$ days) and HGS simulated top 25 cm SM and the ubRMSD between the SMAP SSM-derived SWI (using $T = 20$ days) and HGS simulated top 100 cm SM, respectively, summarized from all SMAP 9-km grids over the study watershed. Within each box, the horizontal segment and the star denote the median and mean of the sample data, respectively; the lower and upper edges of the box indicate the 25th and 75th percentiles, respectively; and the bottom and top ends of the whiskers denote the 5th and 95th percentiles, respectively. (b) and (c): Similar to (a), but for the correlation coefficient $R$ and anomaly $R$, respectively.

Further, the SWI time series are calculated from the watershed averaged SMAP SSM series (using $T = 15$ days and $T = 20$ days, respectively) and are then compared with the watershed averaged HGS SM (top 25 cm and top 100 cm, respectively) (Fig. 9). The SWI time series represent the simulated SM variability in the two soil layers very well, with an $R$ value close to 0.9.  Unsurprisingly, for both soil depth intervals (top 25 cm and top 100 cm), the simulated SM showed a higher correlation with the SWI than with the SMAP SSM (Fig. 9).

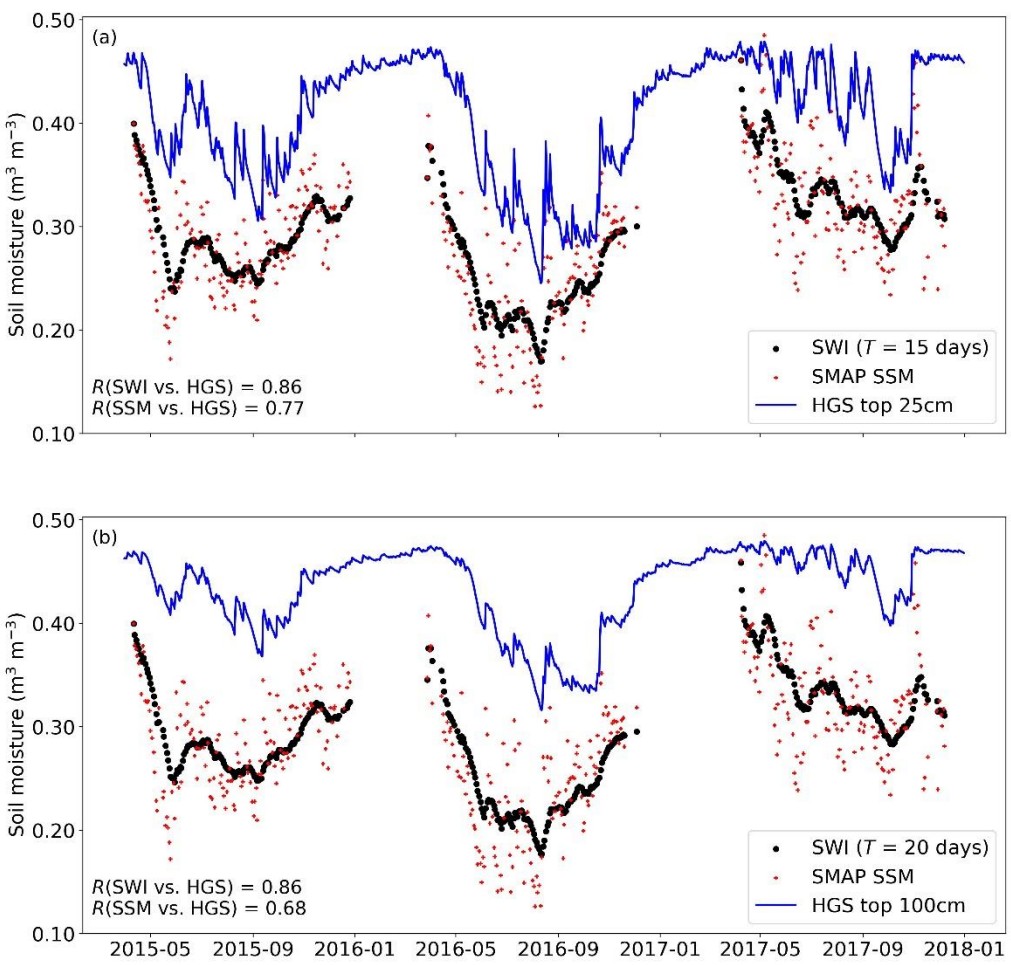

**Figure 9:** (a) Watershed-scale time series of SMAP SSM, SMAP SSM-derived SWI (using *T*= 15 days), and HGS simulated top 25 cm SM. (b) Watershed-scale time series of SMAP SSM, SMAP SSM-derived SWI (using *T*= 20 days), and HGS simulated top 100 cm SM. In each plot, the correlation *R* between SWI (or SMAP SSM) and simulated SM is provided.

Fig. 10 presents the comparison of variations in the HGS simulated watershed integrated GW storage and the watershed scale SWI in the 0-100 cm soil (using *T* = 20 days). The SWI and GW storage share very similar temporal variations, with their best correlation occurring at a time lag of about 2 days. This may demonstrate the potential of SWI to predict the day-to-day variations in GW.

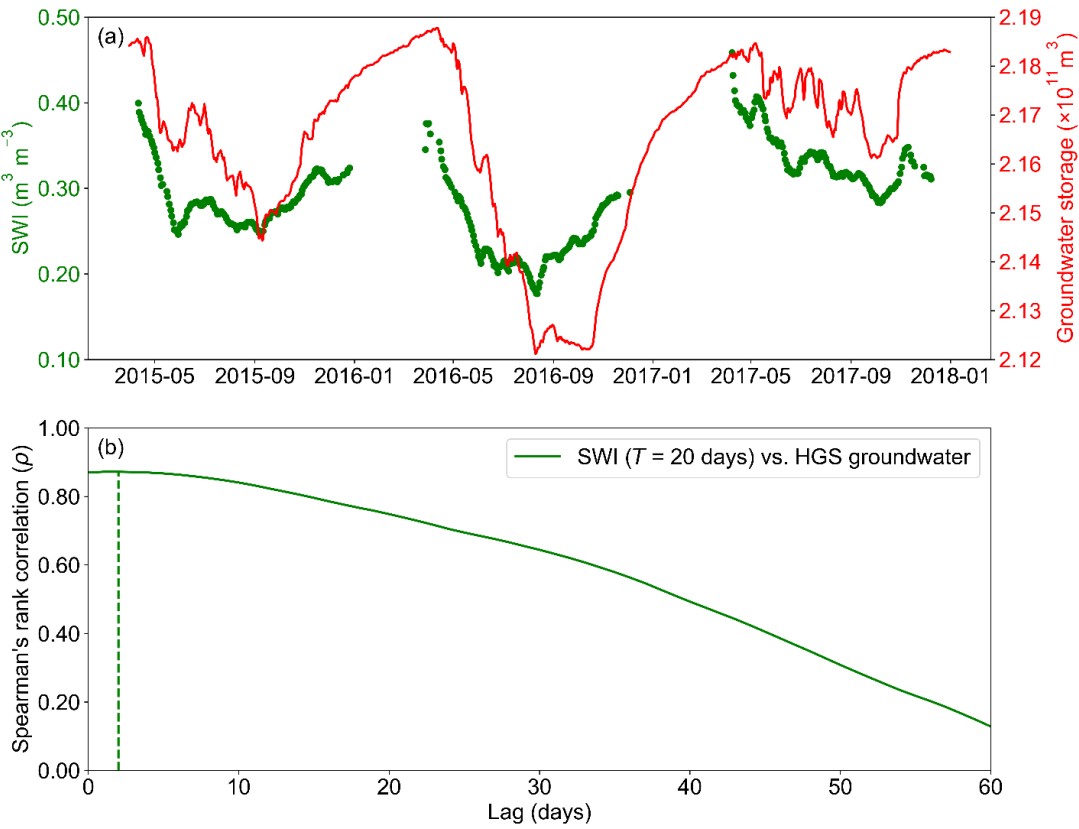

**Figure 10:** (a) Daily spaced time series of watershed-scale SWI (using $T$=20 days) and HGS simulated watershed integrated GW storage. (b) Spearman's rank correlations between the two time series in (a) for time lag ranging from 0 to 60 days. Positive lags indicate that the SWI leads the GW. In (b), the vertical dashed line indicates the location of the time lag leading to the maximum Spearman's rank correlation between the two series.

## 5.2 Identification of Optimal Characteristic Time Length $T_{opt}$

Although the optimal characteristic time length $T_{opt}$ for SWI estimation has been investigated for many regions across the world (e.g. Bouaziz et al., 2020; Ceballos et al., 2005; Tian et al., 2020; Wagner et al., 1999; Zhang et al., 2017), Canadian agricultural watersheds are typically underrepresented in this regard. In this part, $T_{opt}$ is identified for the study watershed by optimally matching variations in SWI and HGS simulated subsurface SM. First, at each SMAP grid, SWI is calculated from the SMAP SSM series for the characteristic time length $T$ varying between 1 and 100 days. Then, Spearman's rank correlations between the SMAP-derived SWI for each value of $T$ and the HGS simulated subsurface SM (from three depth intervals: 0–25 cm, 0–50 cm, and 0–100 cm) are calculated. For each depth interval, the $T$ value corresponding to the highest Spearman's rank correlation is defined as the optimal $T_{opt}$. The optimal $T_{opt}$ (in days) and the associated highest Spearman's rank correlations across all SMAP 9-km grids are provided in Fig. A10. A comparison between Fig. A10 and the soil map

(Fig. 1c) indicates that the spatial variability of $T_{opt}$ is impacted by the soil texture. $T_{opt}$ is typically longer for the landscape with poorly (e.g., Bearbrook) or imperfectly (e.g., Achigan) drained soils than for regions with well drained soils (e.g., Farmington). Table A2 shows the average $T_{opt}$ for the six major soils over the study watershed. For each soil, the averaged $T_{opt}$ is calculated using the 9-km SMAP grids dominated by the soil texture (the soils of Organic and Morrisburg are not calculated and included in the table due to their insufficient sample grids). Clearly, the spatial variability of $T_{opt}$ is strongly related to the soil drainage class. For the three depth intervals: 0–25 cm, 0–50 cm, and 0–100 cm layers, on average, $T_{opt}$ exceeds 20 days, 24 days, and 30 days, respectively, in regions with imperfect or poor soil drainage, while the $T_{opt}$ values are reduced to below 18 days, 21 days and 28 days, respectively, for the well drained soils.

The frequency distribution of $T_{opt}$ at the 9-km grid scale is provided in Fig. A11, while Fig. 11a presents the Spearman's rank correlations (the 5th to 95th percentiles from all SMAP grids over the study watershed) between the HGS simulated SM (0–25 cm, 0–50 cm, and 0–100 cm depths, respectively) and the SWI using $T$ from 1 to 100 days. Across the SNW, the 9-km grid scale $T_{opt}$ ranges largely from 14 to 26 days (Fig. A11a) with a median of 21 days (Fig. 11a) for the 0-25 cm layer, from 20 to 32 days (Fig. A11b) with a median of 24 days (Fig. 11a) for the 0-50 cm layer, and from 26 to 43 days for the 0-100 cm (Fig. A11c) with a median of 31 days (Fig. 11a) for the 0-100 cm layer. On average, $T_{opt}$ increases with the depth in the soil profile, which agrees with previous studies (e.g., Wagner et al., 1999, Tian et al., 2020; Zhang et al., 2017).

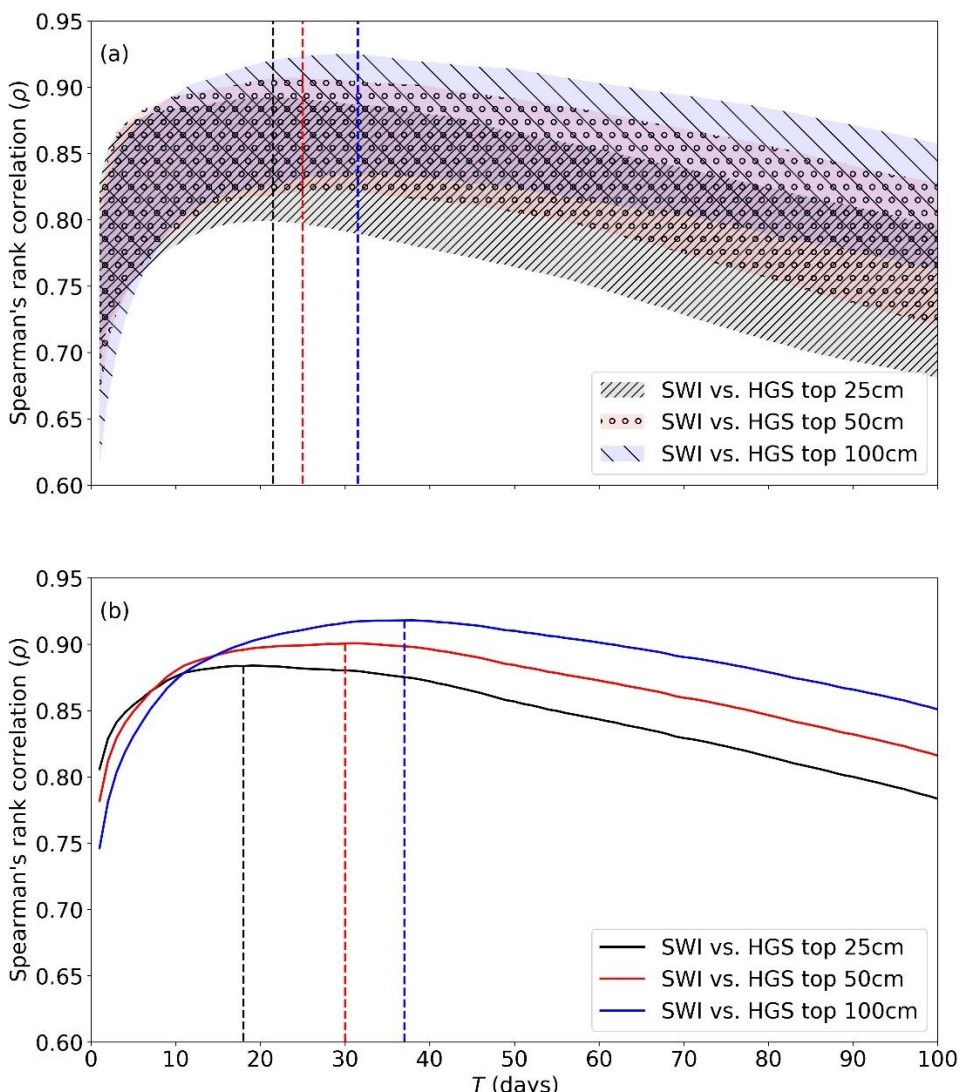


**Figure 11:** (a) Spearman's rank correlations coefficients between the time series (9-km grid scale) of HGS simulated SM (from the 0–25 cm, 0–50 cm, and 0–100 cm layers, respectively) and SMAP SSM-derived SWI for the characteristic time length $T$ ranging from 1 to 100 days. For each layer, the shaded band represents the 5th to 95th percentiles of the results from all SMAP grids within the study watershed with a vertical dashed line indicating the location of the optimal $T_{opt}$ median from all SMAP grids. (b) Spearman's rank correlation
coefficients between the time series (watershed scale) of HGS simulated SM (from the 0–25 cm, 0–50 cm, and 0–100 cm layers, respectively) and SMAP SSM-derived SWI for $T$ ranging from 1 to 100 days. For each layer, the vertical dashed line indicates the location of $T_{opt}$ at the watershed scale.

The maximum Spearman's rank correlations between SWI and simulated subsurface SM (0–25 cm, 0–50 cm, and 0–100 cm

layer) typically exceed 0.8 at the 9-km scale (Figs. A10b,d, and f). By comparing Fig. A10 and the land cover map (Fig. 1b), the forested area typically shows higher maximum Spearman's rank correlations between the SWI and simulated subsurface

SM than the agricultural fields, which again can be at least partially related to the absence of discretely resolved agricultural tile drainage in the model (see section 6.3).

Fig. 11b provides Spearman's rank correlations between watershed averaged HGS subsurface SM and the SWI estimated from watershed averaged SMAP SSM for $T$ varying from 1 to 100 days. The watershed-scale $T_{opt}$ is about 19 days, 30 days, and 38 days for the 0–25 cm, 0–50 cm, and 0–100 cm layers, respectively, showing a clear increase of $T_{opt}$ with increased soil depth (Fig. 11b). Across the three layers, the watershed-scale $T_{opt}$ falls within the range of the most frequently occurring $T_{opt}$ at the 9-km scale (14 to 26 days for the 0-25 cm layer, 20 to 32 days for the 0-50 cm layer, and 26 to 43 days for the 0-

100 cm layer as indicated in Fig. A11), indicating no significant change in $T_{opt}$ across the two spatial scales.

In addition, note that at both spatial scales and across the three layers, the correlations between the calculated SWI and modeled SM are very strong for a range of $T$ values surrounding $T_{opt}$. For example, the simulated 0-100 cm SM shows a correlation > 0.9 with SWI for $T$ ranging from 19 to 60 days at the watershed scale (Fig. 11b), while a correlation > 0.8 can

be obtained between the simulated 0-100 cm SM and SWI for $T$ ranging from 12 to 68 days at the 9-km scale (Fig. 11a). The selected model-independent value of $T = 20$ days for the 0-100 cm layer SWI (section 5.1) falls within both $T$ ranges and is therefore suitable for the 0-100 cm SWI estimation at both spatial scales (Figs. 8 and 9). Similarly, the selected model-independent value of $T = 15$ days for the 0-25 cm layer SWI is also suitable for both spatial scales.

## 6. Discussion

**6.1 Novelty and Improved Understanding of Near Surface–Subsurface Water Interaction**

This study quantified the potential of using SMAP SSM variability to predict subsurface water dynamics using two independent analysis approaches. The first approach is based upon the time lagged cross-correlation in SM variations between the near surface and deeper soil layers (e.g., Mahmood and Hubbard, 2007; Mahmood et al., 2012; Wu et al., 2002),

which can be used to quantify if the subsurface SM variability could be approximated by delaying the temporal variations in satellite/SMAP SSM. The second approach focuses upon the SWI and optimal characteristic time length estimation, which investigates if the subsurface water content variability can be estimated by smoothing the satellite/SMAP SSM time series with an exponential filter (e.g., Bouaziz et al., 2020; Ceballos et al., 2005; Ford et al., 2014; Paulik et al., 2014; Tian et al., 2020; Wagner et al., 1999). Either analysis approach can be independently used to evaluate the linkage between the

SMAP/satellite SSM variability and the deeper subsurface water content fluctuations. Both approaches indicate that the SMAP/satellite SSM variability is strongly linked to the deeper subsurface water content fluctuations and can be used to predict or infer subsurface SM and groundwater variability. Both the optimal time lag (for the delaying method) and the

optimal characteristic time length (for the smoothing method) typically increase with the soil depth and are mainly impacted by the soil drainage properties.


The novelty and advances provided by the study herein are as follows. Firstly, there is growing recognition that high resolution integrated surface water-soil moisture-groundwater modelling and forecasting is crucial for landscape scale water resource management (e.g., Simmons et al., 2020 and references therein). However, the assessment of large (i.e., watershed to river basin) scale, high resolution integrated hydrologic simulations is often difficult due to a lack of spatially distributed

observational information. This study attempts to fill this gap by presenting state-of-the-art satellite (SMAP) SM products as a tool for evaluating integrated hydrologic simulations. The investigation indicates that the SMAP product and the fully-integrated hydrologic model simulations are matched very well in terms of the near surface (top few centimeters) SM variability at both the 9-km scale and the watershed scale. Further, the simulated deeper subsurface SM and GW storage fluctuation is lagged and smoothed in relation to the SSM variability captured by SMAP. The quantified connections

between satellite measurements and modelling results demonstrate the capability of the fully-integrated hydrologic model to reproduce water content in the variably-saturated subsurface domain at a spatial scale that aligns with SMAP cell size. The application of SMAP towards high-resolution fully-integrated surface water-groundwater simulations expands upon previous inter-comparisons of satellite SM and simulations produced by land surface models (e.g., Al-Yaari et al., 2014; Dorigo et al., 2010; Draper et al. 2013; Parrens et al., 2012) or lumped models (e.g., Bouaziz et al., 2020).


Secondly, the study of coupling between near surface-subsurface water fluctuations was extended to the saturated zone (GW) and investigated at multiple spatial scales in this work. In previous work, vertical coupling analyses typically included only the unsaturated zone (surface SM versus root zone SM) for point-scale or small catchments. For example, Mahmood and Hubbard (2007) and Mahmood et al. (2012) quantified the coupling and time lags between near-surface and root zone

SM dynamics at the point or field scale in the US state of Nebraska and suggested that the strength of the coupling was subject to soil type, land use type, and climate, with the temporal delay ranging from several days to a few months. Herein, the high-resolution integrated model simulations enabled an investigation on the vertical coupling and response time differences between dynamics of satellite SM and subsurface water in both unsaturated and saturated zones (i.e., variably-saturated subsurface water) at both the 9-km grid scale and the watershed scale. Results from the two spatial scales showed

consistent variation in vertical coupling and response time across different layers. At both scales, root zone SM and GW fluctuation can be approximated by shifting the SMAP SSM time sequences forward by a soil property dependent optimal time length that increases with subsurface depth. Over the SNW, where poorly or imperfectly drained soils dominate the agricultural regions, the optimal time lag (relative to the SSM variability) is about 1 day for the 25–50 cm SM, about 6 days for the 50–100 cm SM, and about 11 days for the GW storage at both scales. These findings have important implications for

exploiting the potential of SMAP (or other satellite) SSM measurements for estimating subsurface water dynamics in deeper unsaturated and saturated zones. In particular, large-scale satellite SSM monitoring could provide a quick approach for

predicting deeper subsurface water storage changes at continental or global scales and alleviate the need for hydrologic modeling in some types of investigations.

Thirdly, this work suggests optimal and appropriate time length $T$ values for satellite-based SWI estimation, and provides insight on linkages between SWI and subsurface water variability in both unsaturated and saturated zones over a representative Canadian agricultural watershed. Since $T_{opt}$ for SWI estimation is dependent on a number of factors, including subsurface depth of interest, soil properties (e.g., Ceballos et al., 2005; de Lange et al., 2008; Wang et al., 2017), climate (e.g., Albergel et al., 2008; Mahmood et al., 2012; Wang et al., 2017), and land cover/land use (Bouaziz et al., 2020;

Mahmood and Hubbard, 2007), characterization of $T_{opt}$ has been extensively studied. As pioneer of the SWI approach, Wagner et al. (1999) recommended a $T_{opt}$ of 15 days and 20 days for the top 20 cm layer and top 100 cm layer, respectively, based on satellite and in situ SM monitoring over Ukraine. Zhang et al. (2017) reported a $T_{opt}$ of 8 days for the 25 cm depth and 49 days for the 75 cm depth based upon in situ measurements in the US state of Oklahoma. Bouaziz et al. (2020) indicated that $T_{opt}$ values varied significantly across different regions; and when using the SMAP SPL3SMP-E SM product

(also used in the present study), $T_{opt}$ values ranged from ~ 2 to 42 days (within the 5% to 95% percentiles) with a median of around 25 days across their sixteen study catchments in France. Tian et al. (2020) obtained a median $T_{opt}$ of 10 days for the top 70 cm layer using SMAP SSM across in situ SM monitoring sites in the Heihe River basin, China. Canadian agricultural areas are typically underrepresented in previous SWI-related studies, and hence the present study helps fill this gap. In this study, the obtained optimal $T_{opt}$ values for the entire root zone (0–100 cm layer) at the watershed scale (~38 days) and for the

majority of the 9-km grid cells (26 to 43 days) over the SNW are similar to those quantified in other agricultural regions (e.g., Bouaziz et al., 2020; Ceballos et al., 2005). The spatial variability of 9-km scale $T_{opt}$ reasonably reflected the impact of soil texture. Note that at both scales (9-km and watershed) there is a range of $T$ values surrounding $T_{opt}$ that produce high correlations between the calculated SWI and modeled subsurface SM. As such, subsurface moisture variability over the SNW can be well represented by the SMAP-derived SWI using the classic $T$ values (15 days and 20 days for the 0–20 cm

and 0–100 cm layers, respectively). The analysis of optimal (and appropriate) time length $T$ values in this study provide important guidance for SWI estimation over Canada and other agricultural regions around the world. Furthermore, GW storage showed a similar but slightly delayed day-to-day variation relative to SMAP-derived SWI in the 0–100 cm layer, which further supports the use of satellite-derived SWI for detecting GW changes over a range of different time scales (e.g., Sutanudjaja et al., 2013).


**6.2 Point-scale analysis**

With the in situ soil moisture measurements at the four RISMA stations, the time lags between the variations of SSM (top 5 cm) and subsurface SM at the point scale are investigated and presented in Fig. A12 (other in situ sites are not used since they do not provide the SSM measurements). The optimal time lag is less than 1 day between the SSM and 20 cm depth SM

at all four RISMA stations, consistent with the vertical coupling between dynamics of satellite SSM and the simulated 0–25 cm SM. Across the four RISMA sites, the optimal time differences between the variations of SSM and the 50 cm SM range from 0 to 5 days (0 day for ON2 and ON6, 1 day for ON5, and 5 days for ON4), which is also comparable to the response time difference (about 2 days in the RISMA region) between satellite SSM and the simulated 25–50 cm SM.

The $T_{opt}$ values for SWI estimation based upon the point scale in situ soil moisture measurements at the four RISMA stations are given in Fig. A13. The point-scale $T_{opt}$ values range from 1 to 12 days (1 day for ON2, 2 days for ON6, 3 days for ON4, and 12 days for ON5) for SWI estimation at 20 cm depth, while the point-scale $T_{opt}$ values for SWI estimation at 50 cm depth are mostly shorter than 12 days (although the ON4 site shows an $T_{opt}$ of about 50 days for SWI estimation at 50 cm depth, the confidence interval for the $T_{opt}$ is expected to be relatively wide since the highest Spearman's rank correlation varies little

over a wide range of $T$ values). Overall, the point-scale $T_{opt}$ values are shorter than those derived from the satellite and model simulated SM for the 9-km grid scale and the watershed-scale. This may indicate that the deeper subsurface layers typically show a quicker response to the near-surface moisture content variability at the point-scale.

## 6.3 Limitations

Numerous modeling studies have demonstrated the influences of tile drains on hydrological behavior in tile-drained agricultural catchments or regions (e.g., De Schepper et al., 2015; Hansen et al., 2013; Que et al., 2015; Rozemeijer et al., 2010; Valayamkunnath et al., 2022). A limitation in the present study is that tile drainage was not explicitly resolved in the HGS model. However, this limitation is unavoidable, due to the extremely complex challenge associated with representing what are effectively a large number of field scale drainage features in a fully-integrated surface water – groundwater model

for a ~ 3900 km$^2$ watershed. While HGS has previously been used to evaluate tile drainage impacts, the focus has been on much smaller (typically < 50 km$^2$) catchments (e.g., Boico et al., 2022; De Schepper et al., 2015; 2017). To quantify the impact of tile drain omission on the study herein, the fraction of tile drains within each SMAP grid cell is calculated and evaluated in the context of the comparison between SMAP and HGS modeling.

Fig. 12 presents scatterplots of tile drain percentage versus calculated performance metrics for the SSM relationship between SMAP and the HGS modeling across all SMAP 9-km cells. The fraction of tile drains shows a statistically significant positive (negative) correlation with the ubRMSD ($R$ and anomaly $R$). Further, Fig. 13a shows a scatterplot of tile drain percentage versus maximum Spearman's rank correlation between SWI and simulated SM in the 0–100 cm layer, with there being a statistically significant decrease in correlation strength as tile drainage increases. All these indicate that the lack of

tile drainage representation impacted the model performance over the tile drained areas, while also explaining the better agreement between model simulations and SMAP (SSM and SWI) in forested areas than over agricultural fields (Figs. 3, A10b, A10d, and A10f).

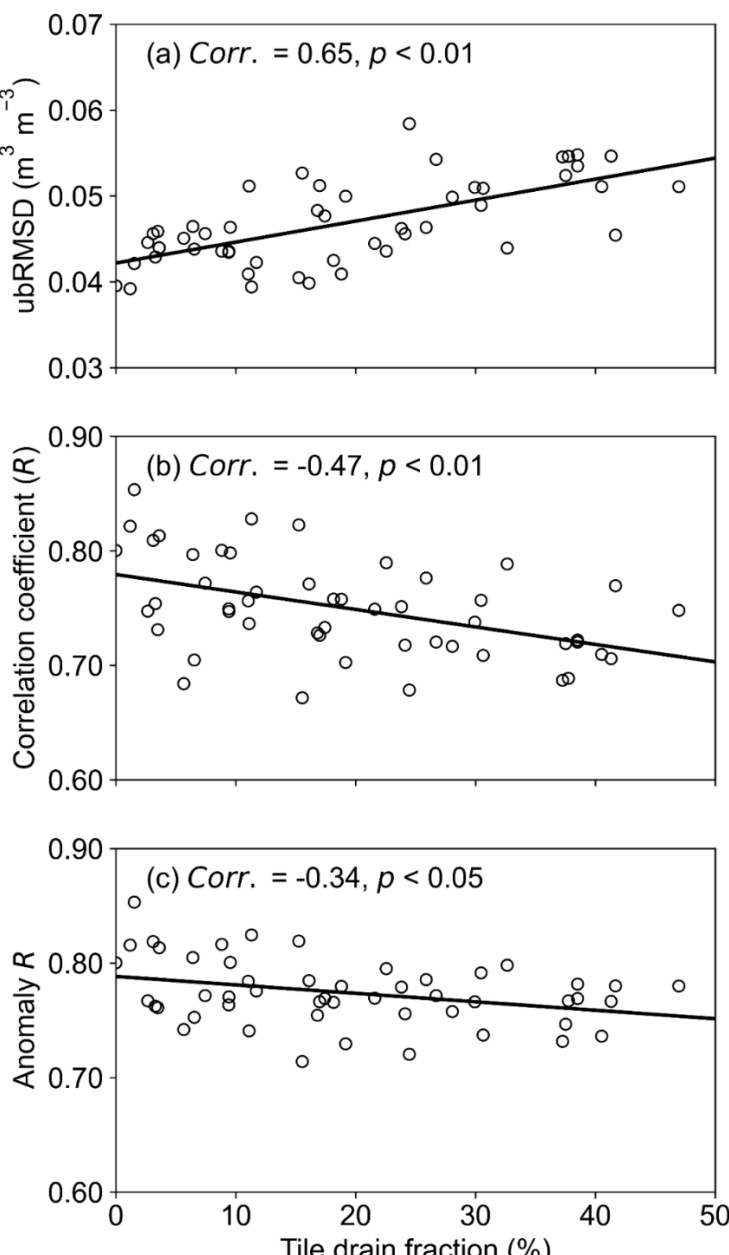

**Figure 12:** Scatterplot of the percent area of tile drains versus (a) ubRMSD, (b) *R*, and (c) anomaly *R*, respectively, across all SMAP grids
within the SNW. The calculated performance metrics (shown in Fig. 3) were for HGS simulated SSM vs SMAP SSM. The Pearson
correlation between tile drain fraction and each performance metric with *p*-value are provided in each plot.

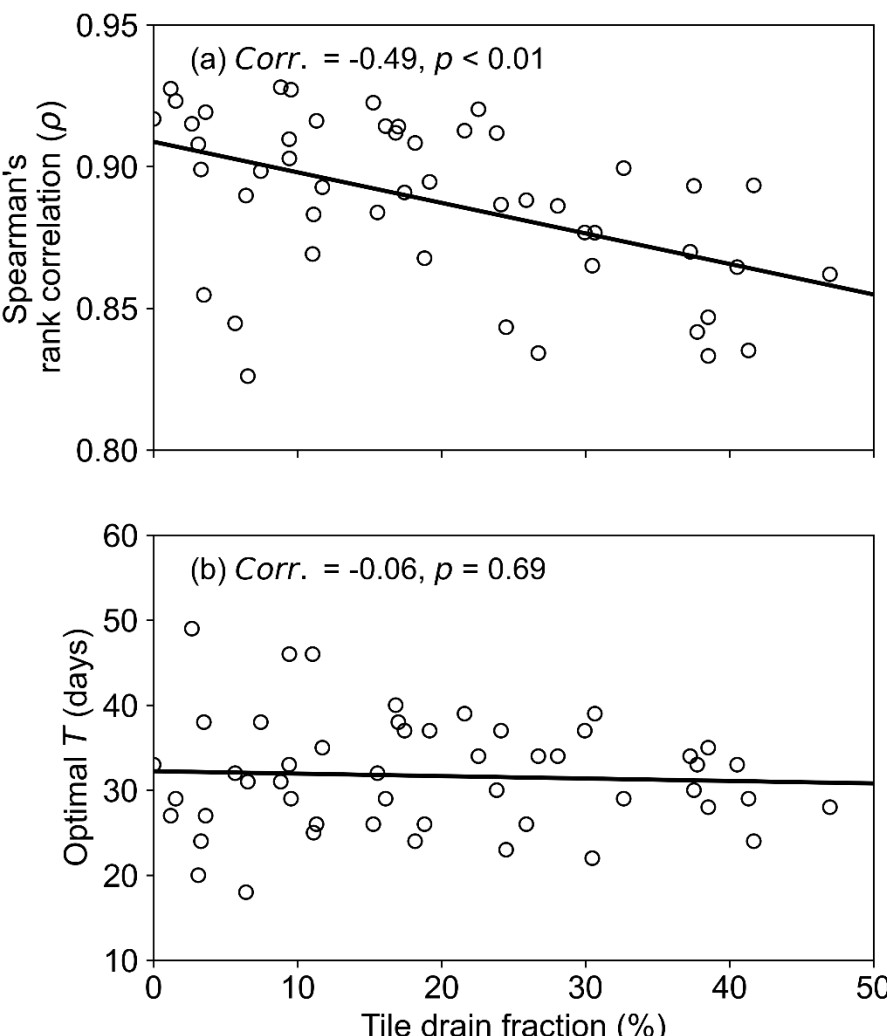

**Figure 13:** Scatterplot of the percent area of tile drains versus (a) the highest Spearman's rank correlation between the SWI series and HGS 0–100 cm SM (as shown in Fig. A10f) and (b) the identified optimal $T_{opt}$ value for the root zone layer (as shown in Fig. A10e) across all SMAP grids. In each plot, the Pearson correlation between the two variables with $p$-value are provided.

However, the tile sensitivity analysis also suggests that the tile drain omission would not negate the findings of the study since it is expected that agreement/linkages between SMAP and HGS modeling would be improved (rather than being discouraged) if tile drainage is explicitly included in the HGS model. Additionally, because the total tile-drained area is only about 25% of the entire watershed and the fraction of tile drains is less than 30% for the majority of SMAP grid cells (Fig. 12), the linkages between SMAP and fully-integrated surface water–groundwater modeling demonstrated within the results of the study are till representative of the dynamic interplay between near surface–subsurface water over the study watershed.

The other limitation of the study is that the presence of tile drainage may impact accurate estimation of SWI over the SNW through modifying the percolation process. However, the impact of this limitation is expected to be marginal for this study given the following reasons. Firstly, the fraction of tile drainage is relatively low (< 30%) for most (80%) of the SMAP 9-km grid cells (Fig. 13). Therefore, the tile drainage would not materially impact the percolation and the SWI estimation for most of the 9-km grids. At the watershed scale, the percentage of total tile-drained area is only about 25% so that the estimation of

watershed-scale SWI should not be significantly influenced either. Secondly, the tile drainage has little impact upon the identified $T_{opt}$, the only control parameter for the SWI estimation. In this study, $T_{opt}$ was identified by matching the variations in the SWI and simulated subsurface SM. Since the tile drainage was not resolved in the model, the identified $T_{opt}$ and the corresponding SWI estimation was not subject to the presence of tile drainage. Fig. 13b provides the scatterplot of the tile drain percentage versus the identified optimal $T_{opt}$ value for the SWI estimation in the 0–100 cm layer. The identified optimal

$T_{opt}$ did not exhibit a statistically significant variation with the fraction of tile drainage.

### 6.4 Other SMAP soil moisture products

In this study, only the SMAP enhanced L3 radiometer 9 km EASE-grid SM (SPL3SMP_E) product (O'Neill et al., 2021) was used. The SMAP/Sentinel-1 L2 Radiometer/Radar SM product (Das et al., 2019; 2020), which can provide higher

spatial resolution (3 km and 1 km) SSM, was not used here because the temporal resolution of the product (~ 12 days) is not appropriate for detecting the time lags between the variations of SSM and subsurface SM. Further, although the SMAP Level-4 (L4) product can provide the surface (0-5 cm) and root-zone (0-100 cm) SM data at 3-h intervals over 9-km EASE-grid (Reichle, et al., 2022), the product is also not suitable for the approaches utilized in this study since the L4 root-zone SM variability is not independent of the SMAP L3/L4 SSM variability. The links between the SMAP SSM and L4 root -zone

SM variations are controlled by the Catchment land surface model and the assimilation system of SMAP brightness temperatures that were used for producing the L4 product. However, note that the SMAP L4 product is in very good agreement with the HGS model simulations, which were used for representing the subsurface water dynamics in this work, in terms of the root zone SM variability (Figure 14A; the absolute bias between them has no impact on the approaches used in this work, which considering only the temporal variations of SM). This further supports the HGS model's application

towards representing the dynamic behavior of subsurface water in this work.

### 7. Conclusion

The inter-comparison and quantified linkage between the two independent data sources: SMAP measurements (SSM and SWI) and HGS fully-integrated surface water – groundwater simulations over a representative agriculture-dominated watershed in eastern Canada led to improved insights into the dynamics of near surface–subsurface water interaction and the

capabilities and approaches of satellite-based SM monitoring and high resolution fully-integrated hydrologic modeling. The

SSM variability is a strong reflection of the deeper subsurface water storage fluctuation, and results support the use of SMAP SSM measurements as indicators and/or predictors of root zone SM and shallow GW storage dynamics. Furthermore, the subsurface SM variability can be well represented by SMAP-derived SWI series, which can also be used to predict shallow GW storage change. The vertical coupling strength and the time scale for water traveling from the near-surface to deeper subsurface did not exhibit statistically significant differences across the two spatial scales of investigation, namely SMAP 9-km grid cell and watershed. The high-resolution fully-integrated hydrologic simulations conducted with the HGS model performed well in terms of reproducing the variably-saturated subsurface water dynamics, although adding the representation of tile drains to the model would further improve the model performance for the tile-drained regions of the subject watershed through the use of remote sensing based SM measurements as validation targets. As satellite SM monitoring continues to evolve, this study has important implications for exploiting the potential of satellite-based SM to predict root zone SM and GW dynamics.

**Data Availability**

The SMAP enhanced L3 radiometer 9 km EASE-grid soil moisture (SPL3SMP_E) version 5 product (O'Neill et al., 2021) and the model output from HydroGeoSphere (HGS) (Aquanty, 2022) were used in the creation of this manuscript. The data are publicly accessible at https://doi.org/10.5281/zenodo.8145252 (Nayak et al., 2023).

**Author contribution**

AN, XX, and SF conceived the study. AN and XX analysed the data and wrote the manuscript draft. SF, OK, AE, DL, HR, and ES contributed to the data curation, analysis and the editing of the manuscript.

**Competing interests**

The authors declare that they have no conflict of interest.

**Acknowledgments**

We acknowledge the support of the Canadian Space Agency (CSA) Grant 21SUESMVAS and the Canada 1 Water project. Thanks go to the NASA National Snow and Ice Data Center Distributed Active Archive Center for providing access to the SMAP soil moisture product. Thanks also go to the South Nation Conservation Authority for providing the tile drain information.

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

**Appendix A**

**Table A1**   Averaged optimal time lags for different soils

| Soils | Soil Drainage Class | Number of samples | Averaged optimal time lag with the 95% confidence intervals (in days) | | | |
|---|---|---|---|---|---|---|
| | | | 0–25 cm SM | 25–50 cm SM | 50–100 cm SM | GW |
| ACHIGAN | Imperfectly drained | 9 | 0, [0, 0] | 1.00, [0.78, 1.22] | 13.0, [11.2, 14.4] | 12.3, [11.8, 13.1] |
| BAINSVILLE | Poorly drained | 5 | 0, [0, 0] | 1.20, [1.00, 1.43] | 3.6, [3.0, 4.3] | 18.8, [14.3, 21.7] |
| BEARBROOK | Poorly drained | 10 | 0, [0, 0] | 1.00, [0.82, 1.15] | 9.6, [7.5, 11.0] | 13.7, [13.5, 14.0] |

| BRANDON | Poorly drained | 14 | 0, [0, 0] | 2.21, [1.50, 2.70] | 5.9, [5.3, 6.6] | 13.6, [13.4, 13.9] |
| FARMINGTON | Well drained | 4 | 0, [0, 0] | 0.75, [0.43, 1.00] | 4.5, [4.0, 4.7] | 7.5, [6.0, 8.0] |
| GRENVILLE | Well drained | 6 | 0, [0, 0] | 0.67, [0.43, 0.86] | 6.0, [5.0, 6.8] | 8.7, [7.2, 10.0] |

**Table A2** Averaged optimal characteristic time length $T_{opt}$ for SWI estimation for different soils

| Soils | Soil Drainage Class | Number of samples | Averaged $T_{opt}$ with the 95% confidence intervals (in days) | | |
|---|---|---|---|---|---|
| | | | 0–25 cm soil layer | 0–50 cm soil layer | 0–100 cm soil layer |
| ACHIGAN | Imperfectly drained | 9 | 25, [23, 26] | 28, [27, 30] | 38, [37, 39] |
| BAINSVILLE | Poorly drained | 5 | 25, [24, 26] | 27, [26, 28] | 30, [29, 31] |
| BEARBROOK | Poorly drained | 10 | 21, [20, 22] | 24, [23, 24] | 32, [31, 33] |
| BRANDON | Poorly drained | 14 | 20, [19, 22] | 27, [26, 28] | 32, [31, 33] |
| FARMINGTON | Well drained | 4 | 18, [17, 20] | 21, [19, 24] | 24, [23, 25] |
| GRENVILLE | Well drained | 6 | 17, [15, 18] | 21, [19, 22] | 28, [27, 29] |

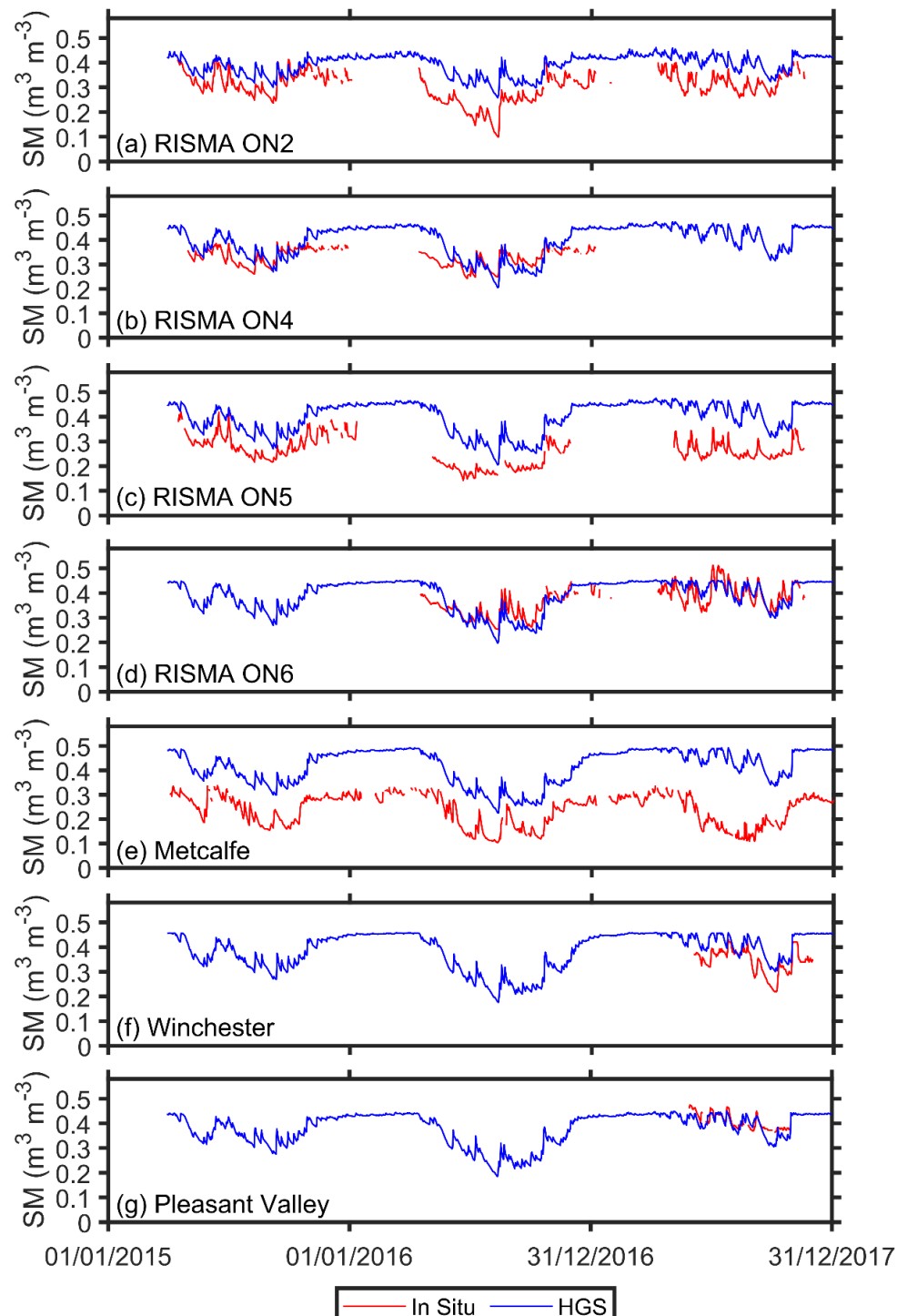

**Figure A1:** The 0-25 cm depth soil moisture time series of HGS versus In Situ at (a) ON2, (b) ON4, (c) ON5, (d) ON6, (e) Metcalfe, (f) Winchester, and (g) Pleasant Valley, respectively.

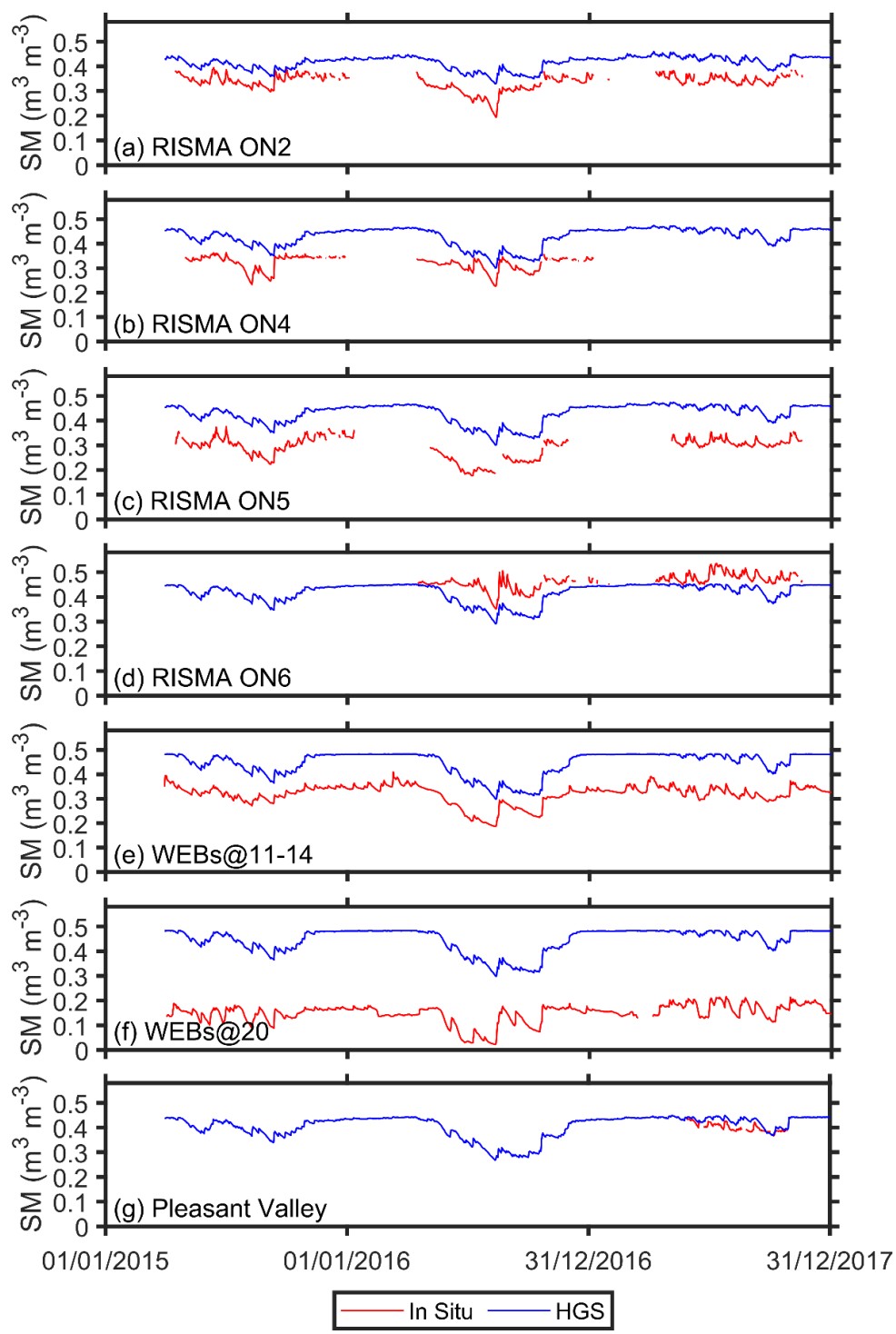

**Figure A2.** The 0-100 cm depth soil moisture time series of HGS versus In Situ at (a) ON2, (b) ON4, (c) ON5, (d) ON6, (e) WEBs@11-14, (f) WEBs@20, and (g) Pleasant Valley, respectively.

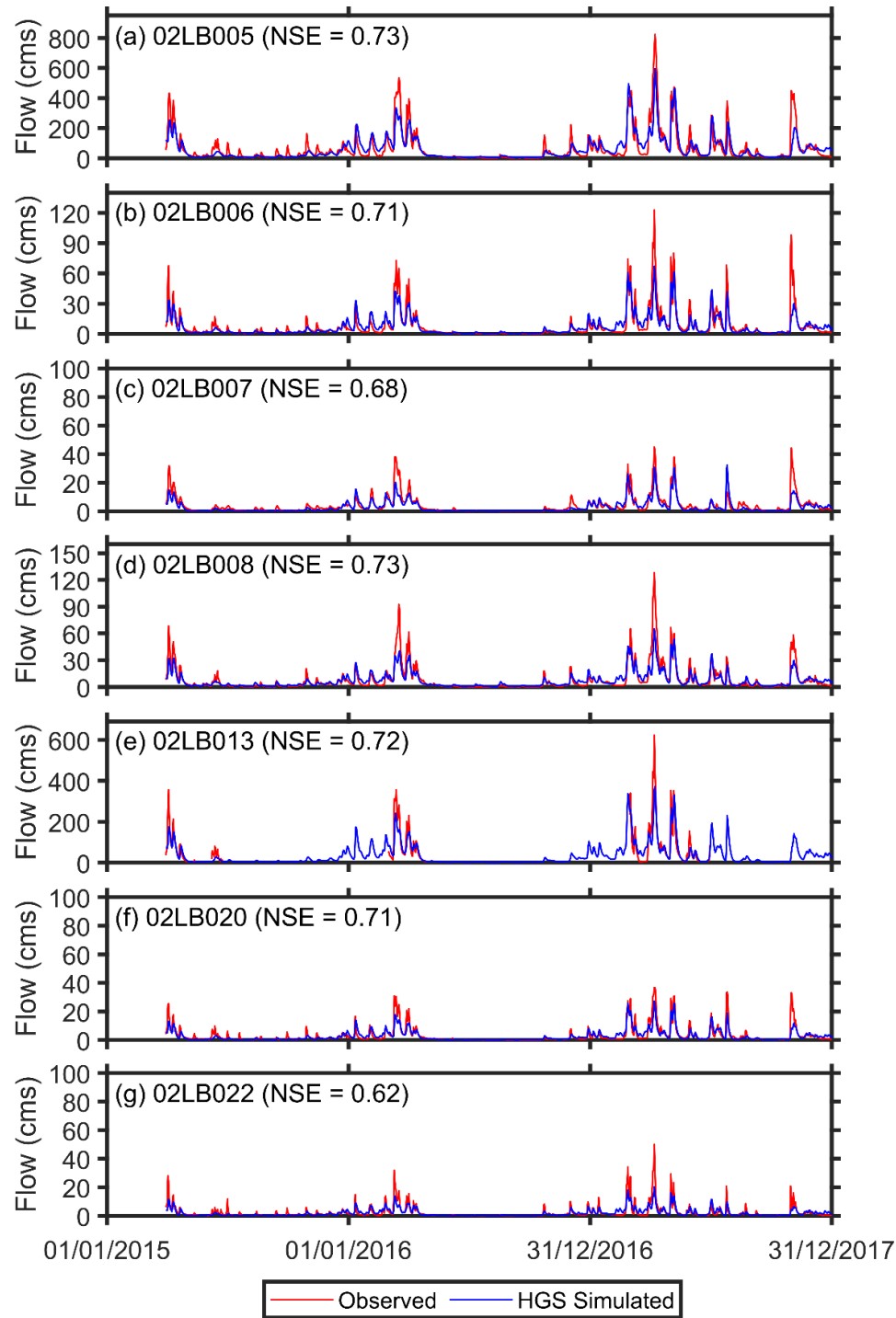

**Figure A3:** Comparison between the observed and HGS simulated hydrographs at the seven gauges: (a) 02LB005, (b) 02LB006, (c) 02LB007, (d) 02LB008, (e) 02LB013, (f) 02LB020, and (g) 02LB022, respectively. The corresponding Nash-Sutcliffe efficiency (NSE) value is provided in each panel.

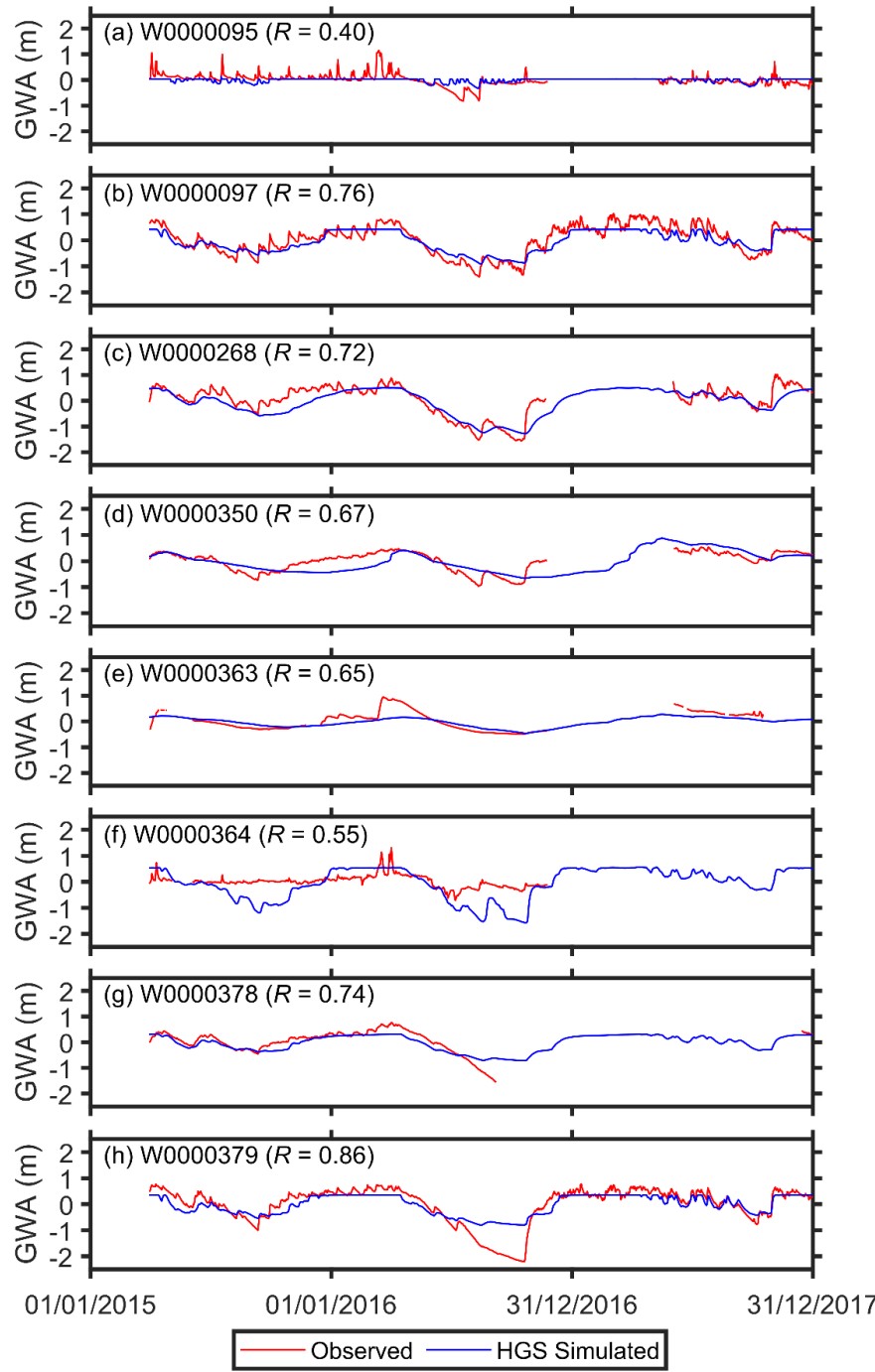

**Figure A4:** Comparison between the observed and HGS simulated groundwater level anomalies (GWA) at the eight GW monitoring wells. The anomalies represent the deviations relative to their respective means over the study period. The corresponding *R* value is provided in each panel.

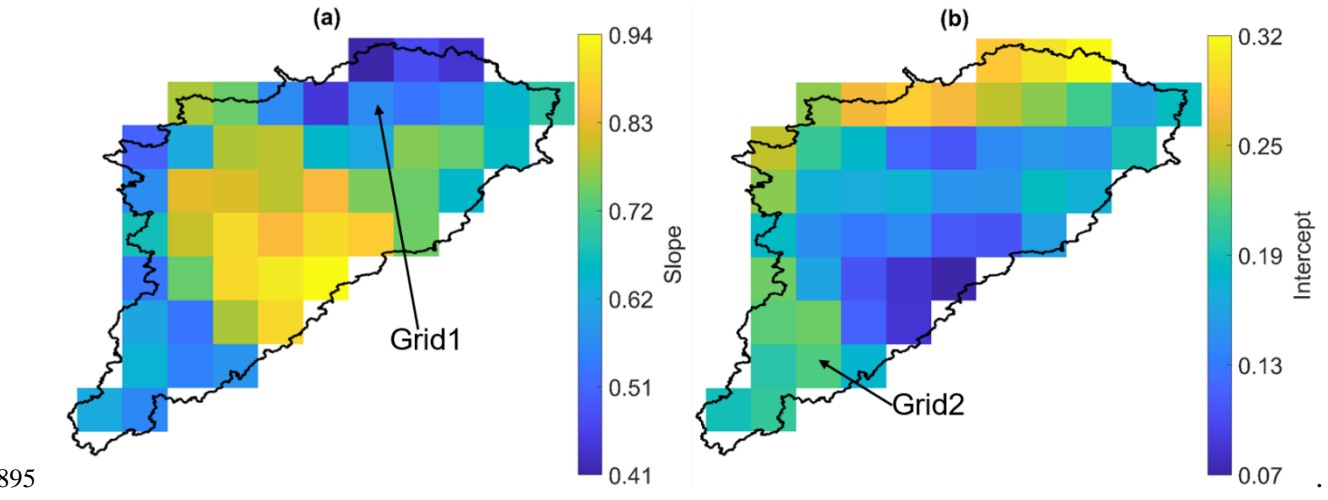


**Figures A5:** (a) Slope and (b) intercept for a linear regression between SMAP SSM (independent variable) versus HGS SSM (dependent variable) across all SMAP grids.

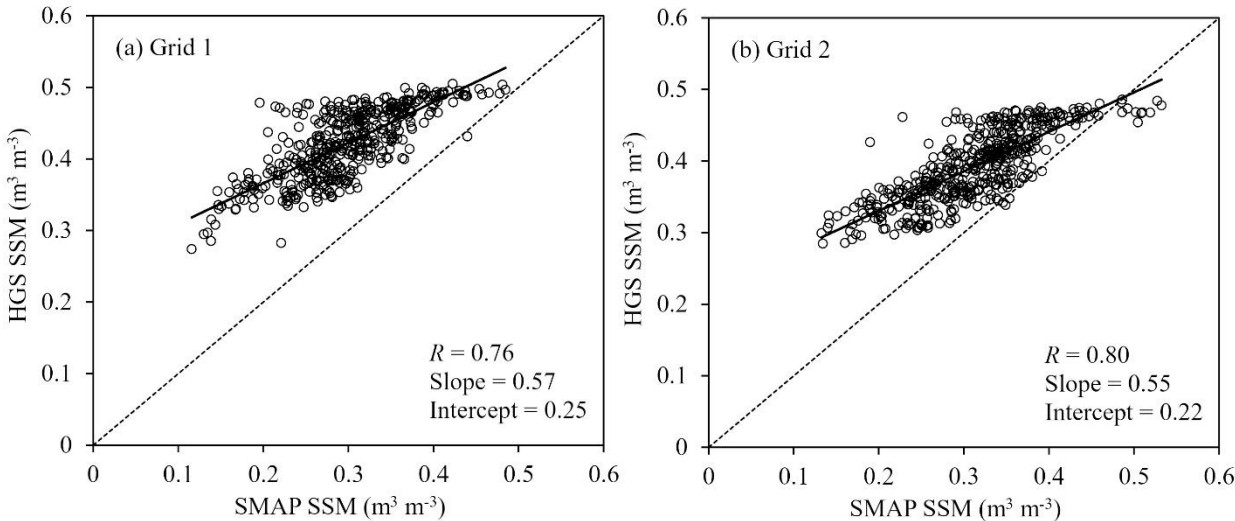


**Figure A6:** Scatterplots between SMAP SSM and HGS SSM for (a) Grid 1 and (b) Grid 2, as shown in Figure A5.

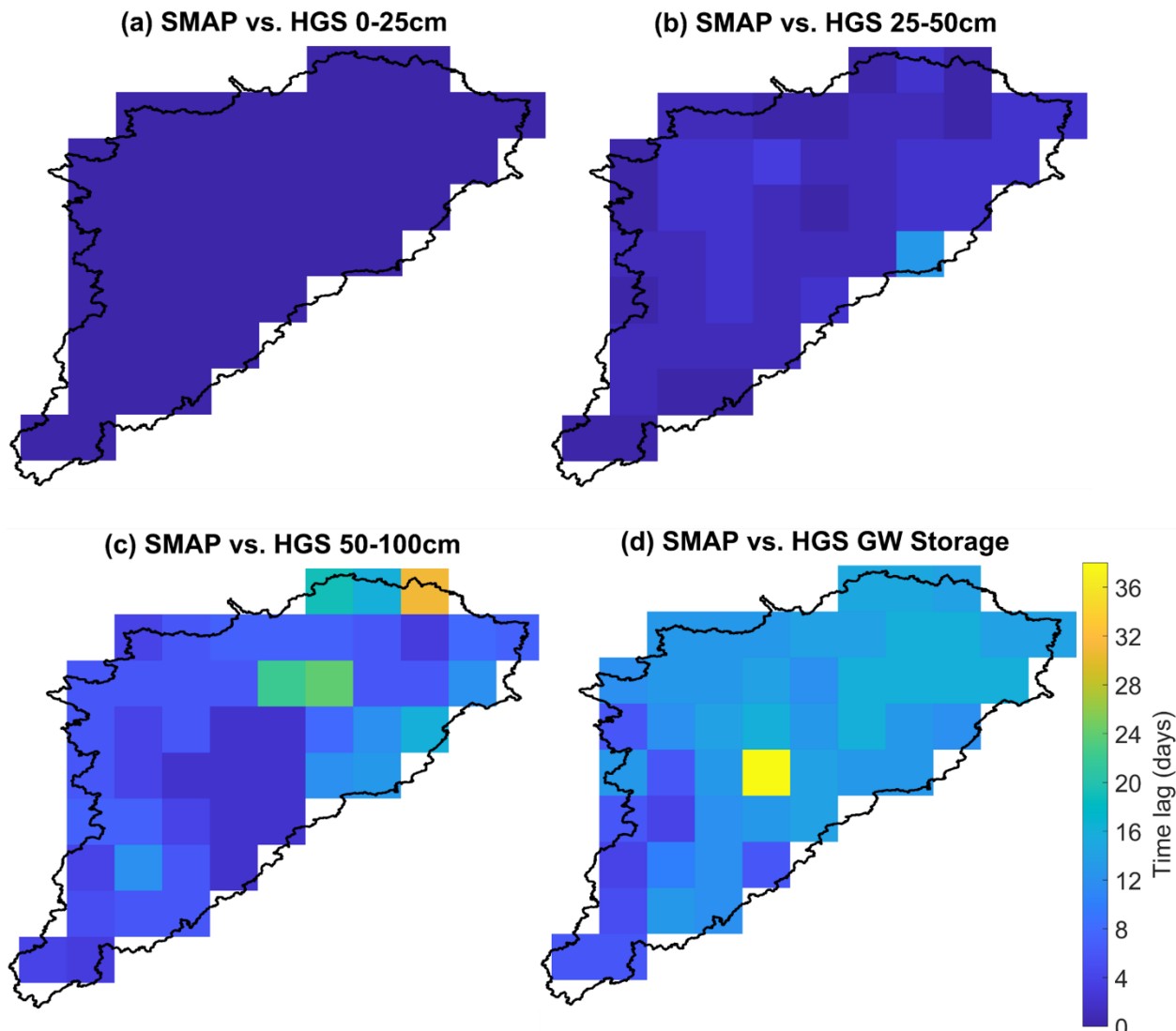

**Figures A7:** Optimal time lag (in days), relative to the SMAP SSM variability, for HGS simulated (a) 0–25 cm SM, (b) 25–50 cm SM, (c) 50–100 cm SM, and (d) GW storage across all SMAP grids.

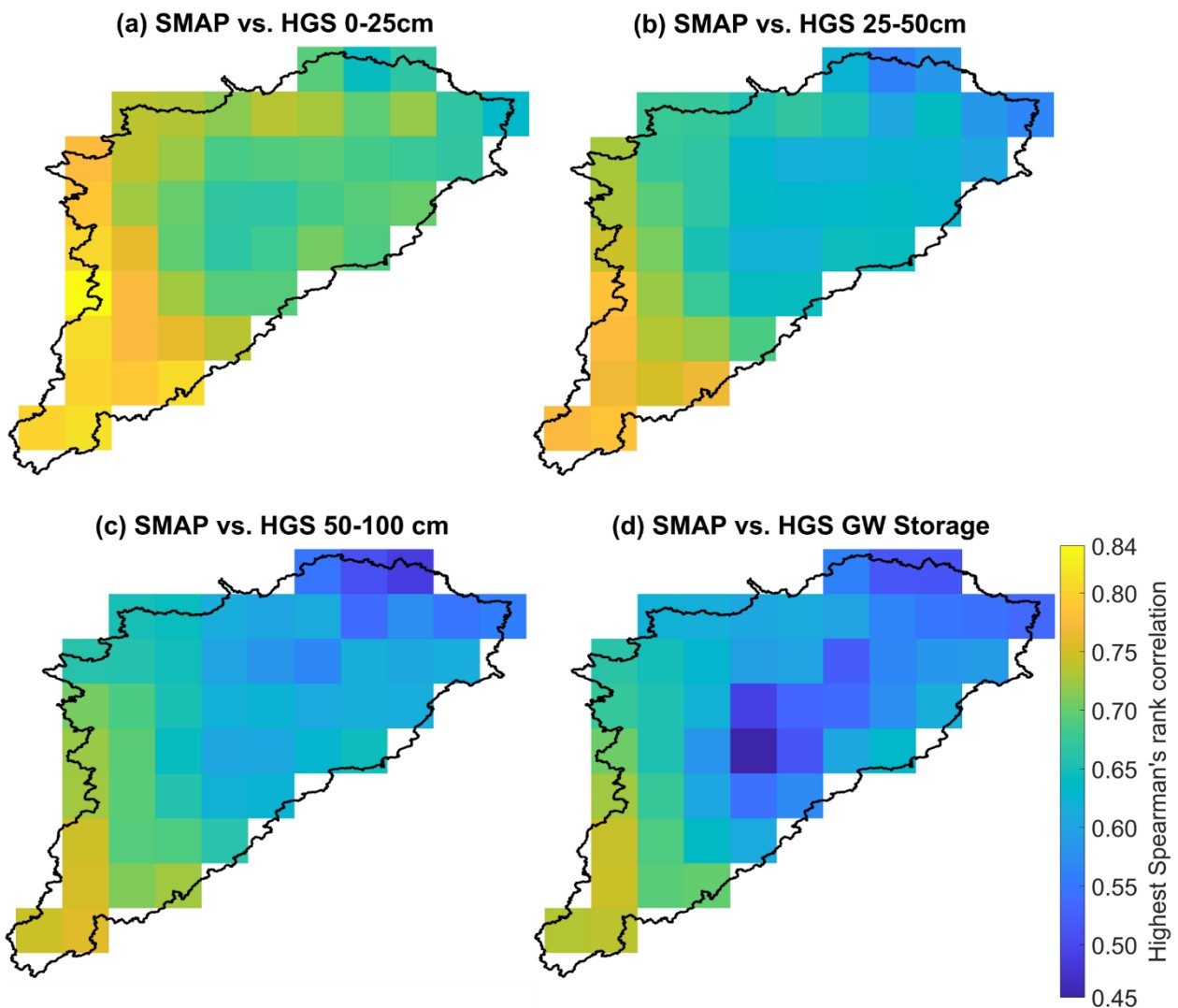

**Figures A8:** Maximum Spearman's rank correlation between SMAP SSM versus HGS simulated (a) 0–25 cm SM, (b) 25–50 cm SM, (c) 50–100 cm SM, and (d) GW storage, respectively, across all SMAP grids.

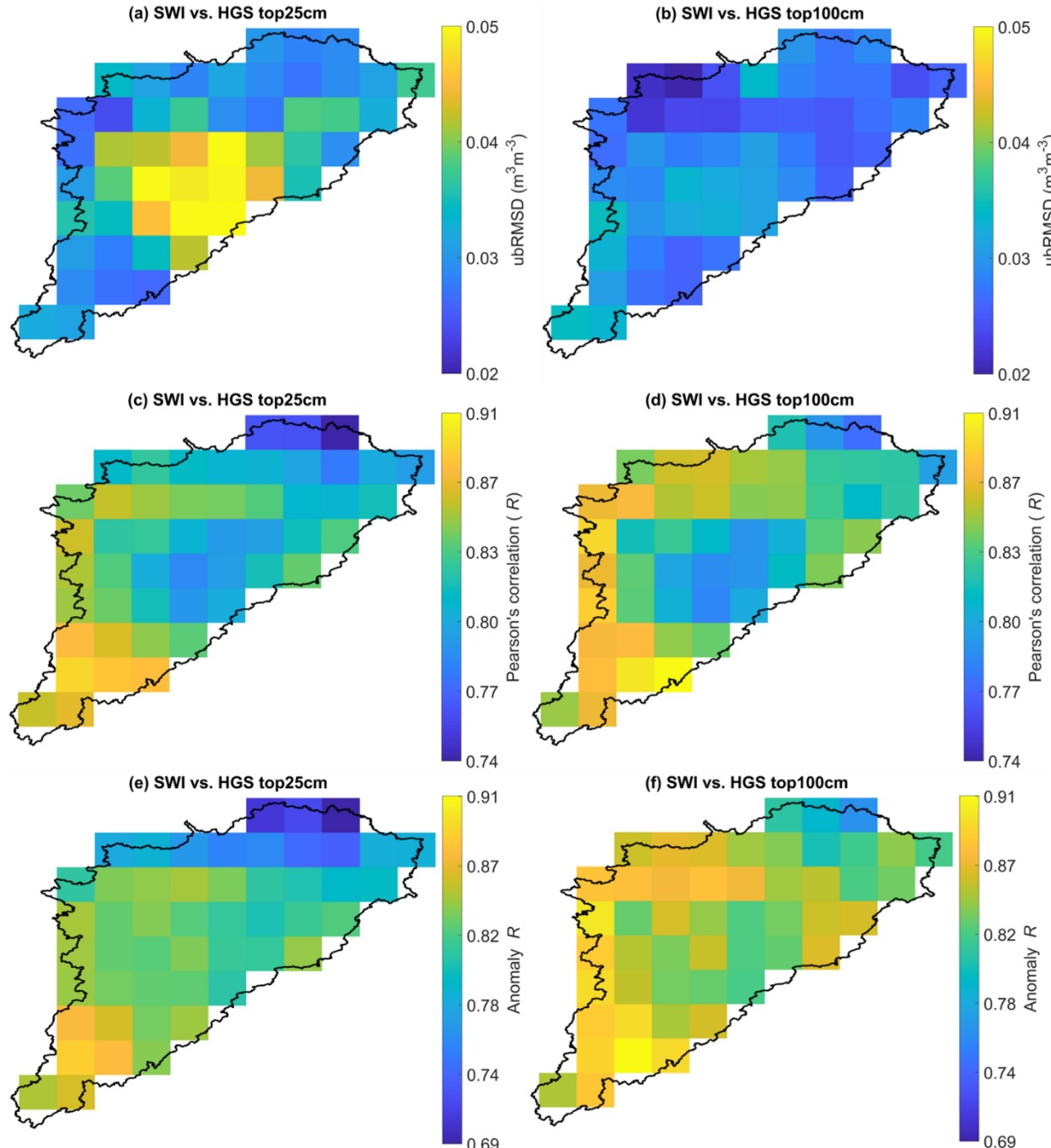


**Figures A9:** Left: (a) ubRMSD, (c) $R$, and (e) anomaly $R$ between the SMAP-derived SWI ($T = 15$ days) and HGS simulated 0–25 cm soil moisture across all SMAP grids over the study watershed. Right: (b) ubRMSD, (d) $R$, and (f) anomaly $R$ between the SMAP derived SWI ($T = 20$ days) and HGS simulated 0–100 cm soil moisture across all SMAP grids over the study watershed.


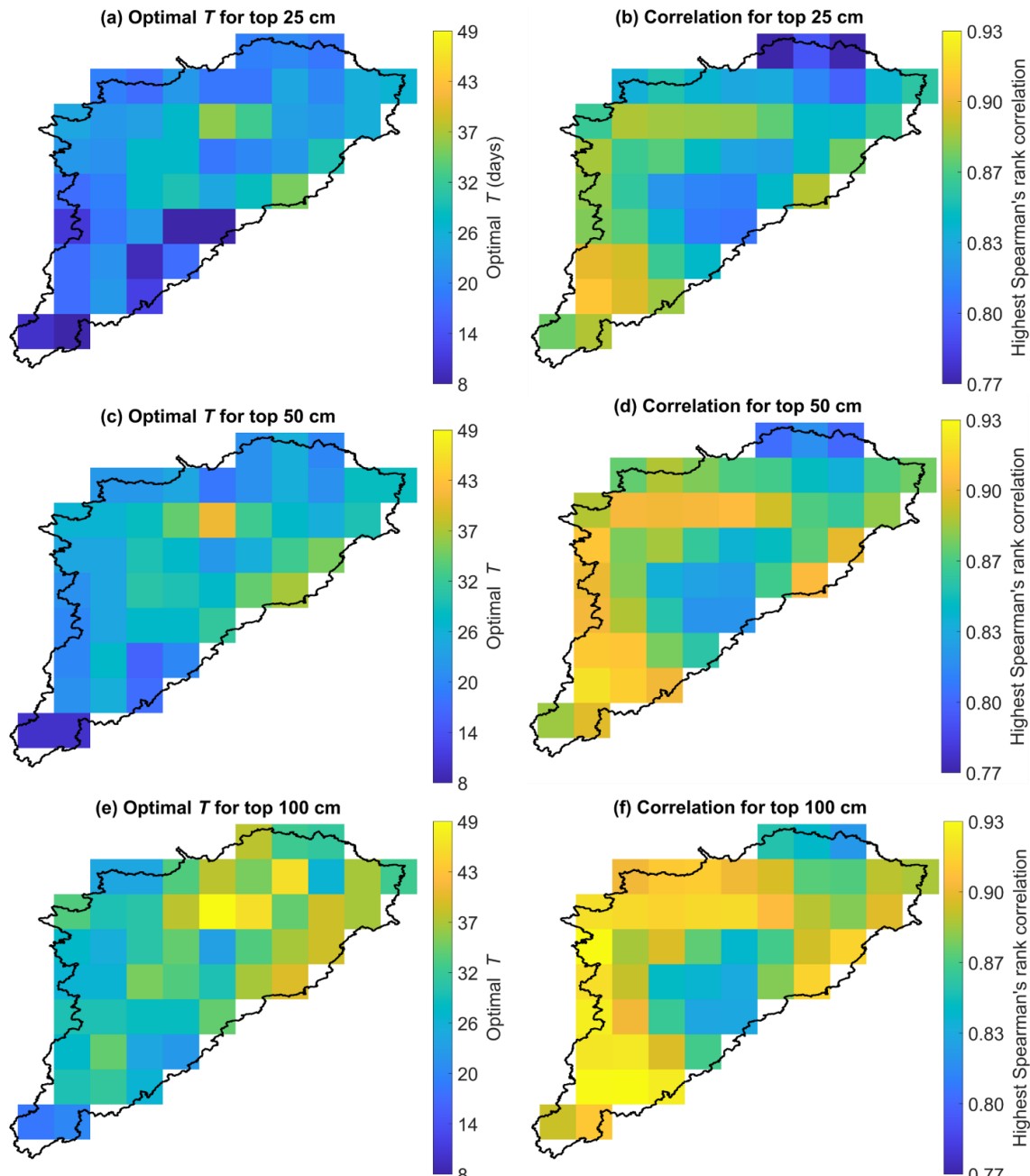

**Figures A10:** Left: Optimal $T_{opt}$ for SWI estimation for (a) 0–25 cm, (c) 0–50 cm, and (e) 0– 100 cm soil layers, respectively. Right: Maximum Spearman's rank correlation between the SWI and simulated soil moisture for (b) 0–25 cm, (d) 0–50 cm, and (f) 0– 100 cm soil layers, respectively.

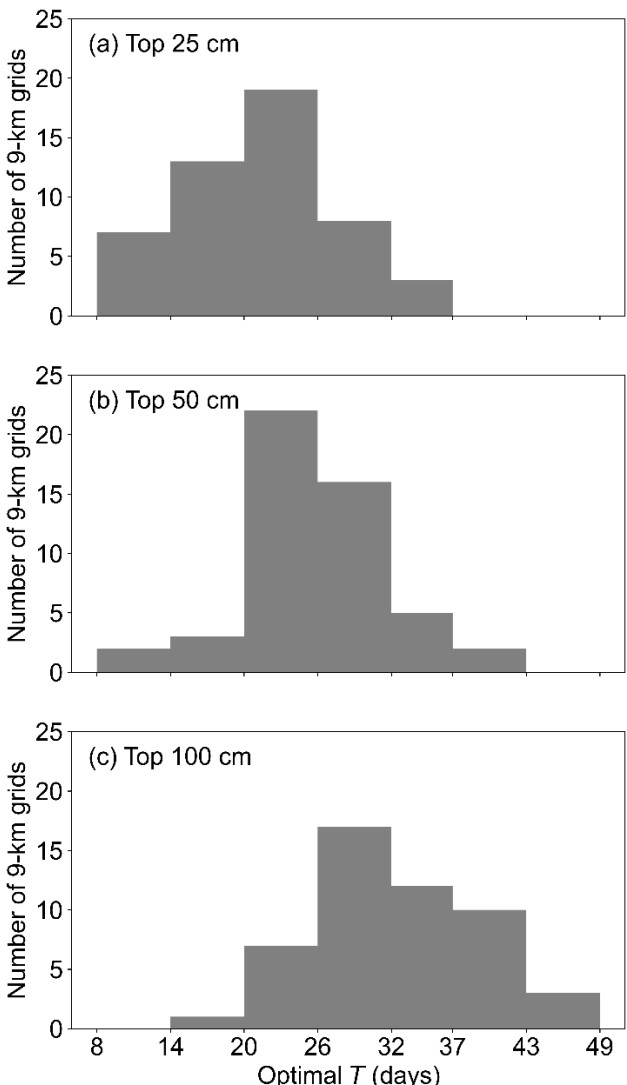

**Figure A11:** Distribution of the optimal $T_{opt}$ at the 9-km grid scale for (a) 0–25 cm, (b) 0–50 cm, and (c) 0–100 cm soil depths, respectively.

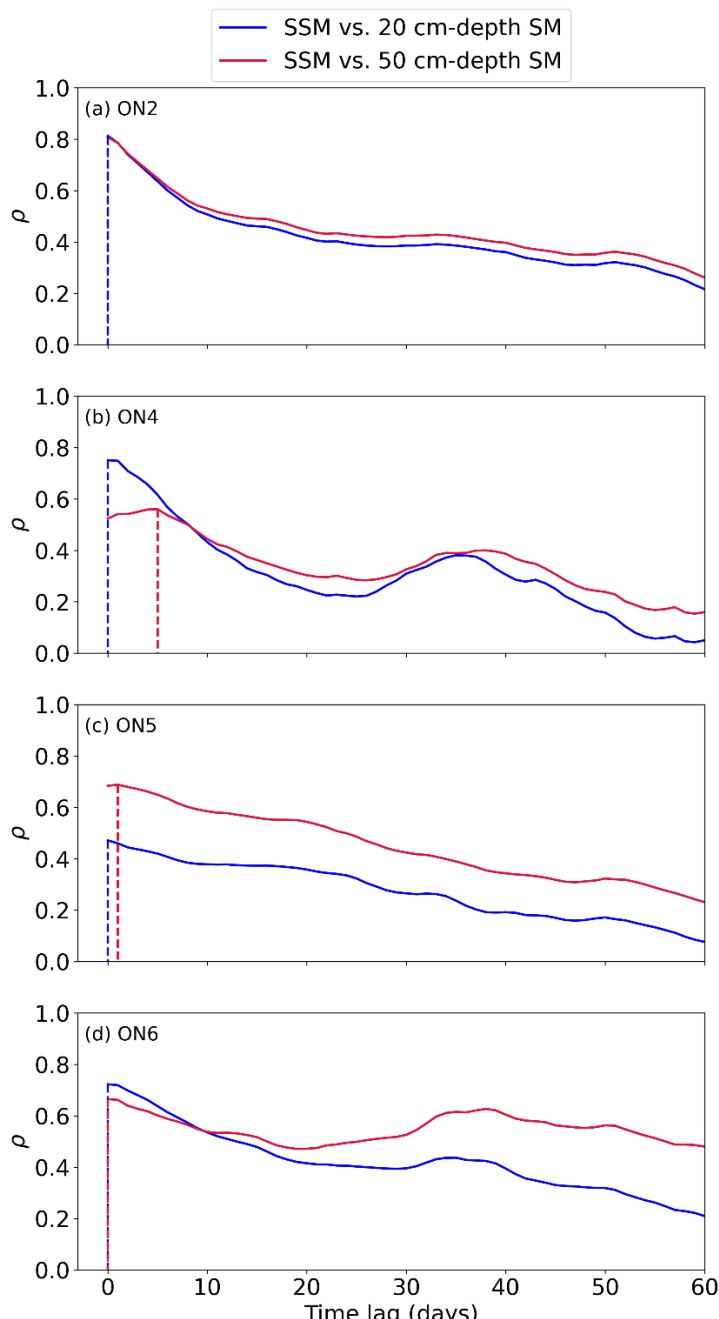

**Figure A12:** Spearman's rank correlation (ρ) between the near-surface (top 5 cm) soil moisture (SSM) and the subsurface (20 cm and 50 cm depths) soil moisture (SM) for the time lag ranging from 0 to 60 days based upon the in situ measurements at the four RISMA stations: (a) ON2, (b) ON4, (c) ON5, and (d) ON6, respectively. Positive lags indicate that the SSM leads the subsurface SM. The vertical dashed line indicates the optimal time lag corresponding to the maximum ρ between the SSM and subsurface SM.


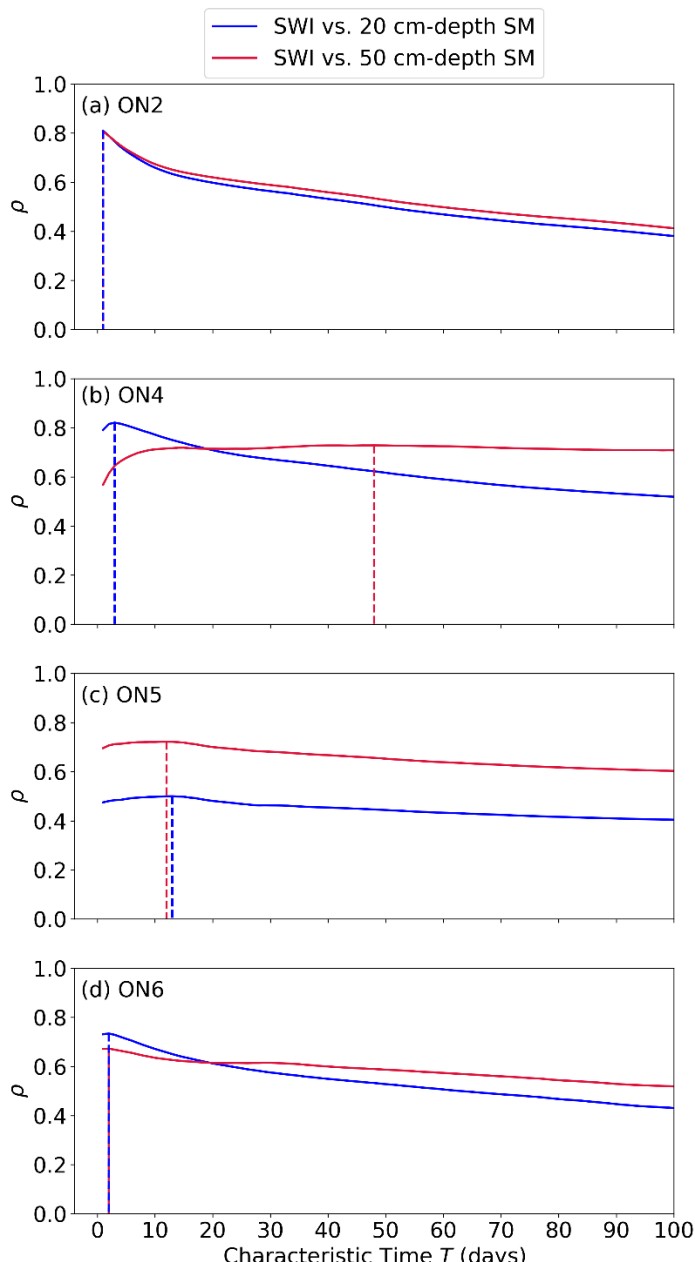

**Figure A13:** Spearman's rank correlation ($\rho$) between the subsurface SM (20 cm and 50 cm depths) and the SSM-derived SWI for the characteristic time $T$ ranging from 1 to 100 days, based upon the in situ measurements at the four RISMA stations: (a) ON2, (b) ON4, (c) ON5, and (d) ON6, respectively. The vertical dashed line indicates the location of the optimal characteristic time $T_{\text{opt}}$ for SWI estimation.

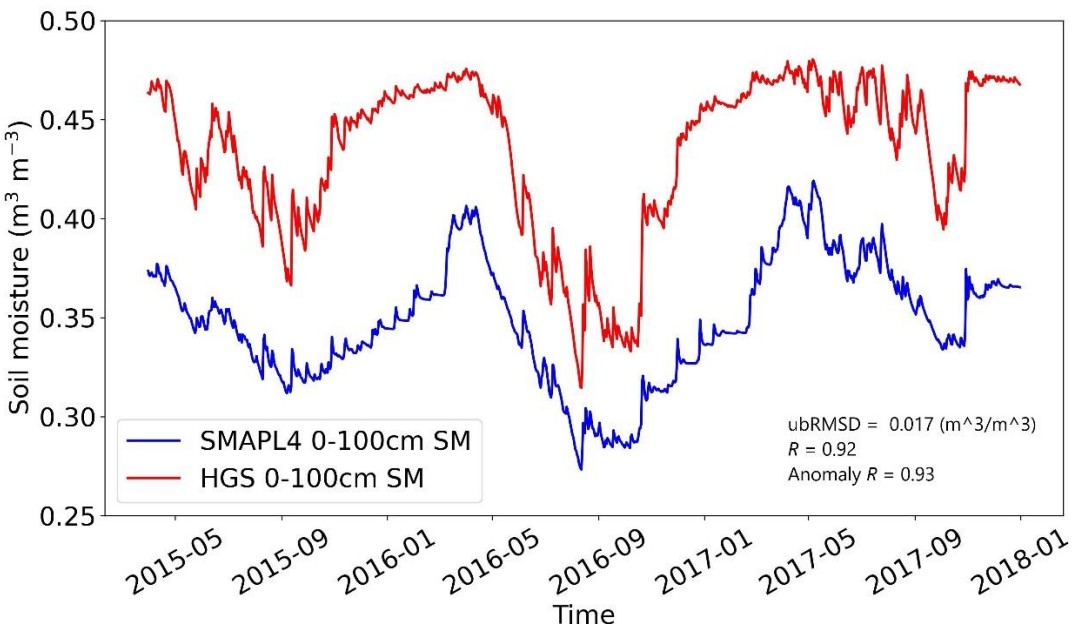

**Figure A14:** Comparison between the SMAP L4 and HGS simulations for the watershed averaged root zone (0-100 cm) SM time series. The error metrics (ubRMSD, *R*, and anomaly *R*) are calculated without considering the data in winter (December to March) where satellite SSM measurements are typically not available.