# Peer review of "Quantifying the potential of using SMAP soil moisture variability to predict subsurface water dynamics"

_Hydrology and Earth System Sciences, 2023_

## Author Comment (AC1)

[Figure]

**Figure S1.** Spearman's rank correlation of the near-surface (top 5 cm) soil moisture with the top 25 cm soil moisture and with the root zone (top 100 cm) soil moisture, respectively, for the time lag ranging from 0 to 60 days based upon the in situ measurements at the four RISMA stations: (a) ON2, (b) ON4, (c) ON5, and (d) ON6. Positive lags indicate that the near-surface soil moisture leads the subsurface soil moisture. For each pair of comparison, a vertical dashed line is provided to indicate the time lag with the maximum Spearman's rank correlation.

---

## Author Response (AR1)

**Author's point-by-point response to the reviewers**

**Editor Evaluations:**

Public justification (visible to the public if the article is accepted and published)**:**
Dear Authors: The manuscript has potential to be published in HESS. Kindly address the comments given by the reviewers and submit for further review. My comments also align with the two reviewers.
This study investigates the potential use of SMAP surface soil moisture (SM) in estimating deep layer SM. However, the connection between the two results sections is weak, and there is a lack of discussion on the relationship between time lag and characteristic time length, which is crucial for deep layer SM estimation. The study also misses an in-depth examination of the factors controlling the characteristic time length and does not address spatial heterogeneity in this parameter. Despite the scarcity of in-situ data, which could provide valuable information, these were not utilized in the analysis. Specific suggestions include moving Table A1 into the main text, addressing mismatches in soil layers in Figures A1 and A2, and providing more details on the spatial heterogeneity in optimal time lags and their comparison with model simulations and SMAP data.

Additional private note (visible to authors and reviewers only):
Dear Authors: The manuscript has potential to be published in HESS. Kindly address the comments given by the reviewers and submit for further review. My comments also align with the two reviewers.
This study investigates the potential use of SMAP surface soil moisture (SM) in estimating deep layer SM. However, the connection between the two results sections is weak, and there is a lack of discussion on the relationship between time lag and characteristic time length, which is crucial for deep layer SM estimation. The study also misses an in-depth examination of the factors controlling the characteristic time length and does not address spatial heterogeneity in this parameter. Despite the scarcity of in-situ data, which could provide valuable information, these were not utilized in the analysis. Specific suggestions include moving Table A1 into the main text, addressing mismatches in soil layers in Figures A1 and A2, and providing more details on the spatial heterogeneity in optimal time lags and their comparison with model simulations and SMAP data.

**Response: Thank you for the remarks. The authors have seriously considered all comments and suggestions from the reviewers and Editor's remarks, and addressed each comment accordingly. Since the Editor's comments align with those from the two reviewers, please see our detailed response to the reviewer comments below.**

**Reply to RC1**

Overall comments:

This is an interesting study discussing the potential use of SMAP surface soil moisture (SM) in estimating deep layer SM. However, I feel the two results sections were not well connected, and there is a lack of linkage between the time lag and characteristic time length.

**Response****: This study used two different (and independent) approaches to quantify the potential of using SMAP SSM variability to predict subsurface water dynamics. The first one focuses upon the time lagged cross-correlation in soil moisture (SM) variations between the near surface and deeper soil layers (e.g., Mahmood and Hubbard, 2007; Mahmood et al., 2012; Wu et al., 2002), which can be used to quantify if the subsurface SM variability could be approximated by delaying the temporal variations in satellite/SMAP near surface SM (SSM). The second result section focuses upon the SWI and optimal characteristic time length estimation, which investigates if the subsurface water content variability can be estimated by smoothing the satellite/SMAP SSM time series with an exponential filter (e.g., Bouaziz et al., 2020; Ceballos et al., 2005; Ford et al., 2014; Paulik et al., 2014; Tian et al., 2020; Wagner et al., 1999).**

**Either approach (the time lag analysis or the SWI characteristic time length approach) can be independently used to quantify if the SMAP SSM variability holds the potential to predict subsurface water dynamics. The use of the two independent approaches would make the conclusions (ie., SMAP/satellite SSM variability is strongly linked to the deeper subsurface water content fluctuations and can be used to predict/infer subsurface SM and groundwater variability) more robust. Since the two time parameters (the time lag and the characteristic time length) are suitable for the independent/different approaches, their calculations are completely independent of each other, i.e., the characteristic time length does not rely upon the time lag and vice versa. Therefore, an explicit (or mathematical) relationship/link between the two time parameters are not applicable (this is also not needed).**

**However, results from the two approaches are well connected in the following aspects: (i) both approaches indicate that the SMAP/satellite SSM variability is strongly linked to the deeper subsurface water content fluctuations and can be used to predict/infer subsurface SM and groundwater variability; (ii) both the optimal time lag (for the delaying method) and the characteristic time length (for the smoothing method) increase with the soil depth, and are also strongly impacted by the soil properties.**

**The connection/relationship between the two result sections is now further clarified in the revised manuscript (Line 494-506):**

*"**This study quantified the potential of using SMAP SSM variability to predict subsurface water dynamics using two independent analysis approaches. The first approach is based upon the time lagged cross-correlation in SM variations between the near surface and deeper soil layers (e.g., Mahmood and Hubbard, 2007; Mahmood et al., 2012; Wu et al., 2002), which can**"*

*be used to quantify if the subsurface SM variability could be approximated by delaying the temporal variations in satellite/SMAP SSM. The second approach focuses upon the SWI and optimal characteristic time length estimation, which investigates if the subsurface water content variability can be estimated by smoothing the satellite/SMAP SSM time series with an exponential filter (e.g., Bouaziz et al., 2020; Ceballos et al., 2005; Ford et al., 2014; Paulik et al., 2014; Tian et al., 2020; Wagner et al., 1999). Either analysis approach can be independently used to evaluate the linkage between the SMAP/satellite SSM variability and the deeper subsurface water content fluctuations. Both approaches indicate that the SMAP/satellite SSM variability is strongly linked to the deeper subsurface water content fluctuations and can be used to predict or infer subsurface SM and groundwater variability. Both the optimal time lag (for the delaying method) and the optimal characteristic time length (for the smoothing method) typically increase with the soil depth and are mainly impacted by the soil drainage properties.*"

There is also lack of an in-depth discussion on the underlying factors controlling the characteristic time length, a key parameter for the deep layer SM estimation. If the goal is to obtain high-resolution and high-quality deep layer SM variations, the authors should at least provide a brief discussion on the spatial heterogeneity in this parameter.

**Response: The soil drainage properties have a key impact on the spatial heterogeneity in the optimal time lag and the optimal characteristic time length. In the revision, the average optimal time lags and the average optimal characteristic time lengths are calculated for different soils over the study watershed** (new Tables A1 and A2, page 38). **The relevant discussion is also added:**

Line 326-334 : "*Table A1 provides the average optimal time lags for the six major soils over the study watershed. For each soil, the averaged optimal time lag is calculated using the 9-km SMAP grids dominated by the soil texture (the Organic and Morrisburg soils are not calculated and included in the table due to their insufficient sample grids). Clearly, the soil drainage has a key impact on the spatial variability of the time lags for deeper layers. The optimal time lag for the 25–50 cm depth is statistically shorter (longer) than 1 day in regions with well drained (imperfectly or poorly drained) soils. Moving to the 50–100 cm depth, on average, the soils of Achigan (imperfectly drained) and Bearbrook (poorly drained) dominated regions experienced the longest optimal time delay (close to or higher than 10 days). Further, the optimal time delay is statistically less (more) than 10 days for the GW system in the areas with good (poor or imperfect) soil drainages.*"

Line 447-452: "*Table A2 shows the average $T_{opt}$ for the six major soils over the study watershed. For each soil, the averaged $T_{opt}$ is calculated using the 9-km SMAP grids dominated by the soil texture (the soils of Organic and Morrisburg are not calculated and included in the table due to their insufficient sample grids). Clearly, the spatial variability of $T_{opt}$ is strongly related to the soil drainage class. For the three depth intervals: 0–25 cm, 0–50 cm, and 0–100 cm layers, on average, $T_{opt}$ exceeds 20 days, 24 days, and 30 days, respectively, in regions with*

***imperfect or poor soil drainage, while the $T_{opt}$ values are reduced to below 18 days, 21 days and 28 days, respectively, for the well drained soils.*** ”

Finally, even though the in-situ data were scarce, they can provide key information on this parameter, but they were not used in the analysis.

**Response: In this revision, we have examined the time lags between the in situ SSM (top 5 cm) and the in situ SM at 25 cm/50 cm depths** (Figure A12, page 49) **and the optimal characteristic time length Topt values for SWI estimation** (Figure A13, page 50) **based upon the point scale in situ soil moisture measurements at the four RISMA stations (other in situ sites are not used since they do not provide the SSM measurements). The relevant discussion is also added at** Line 568-584**:**

“***6.2 Point-scale analysis***

***With the in situ soil moisture measurements at the four RISMA stations, the time lags between the variations of SSM (top 5 cm) and subsurface SM at the point scale are investigated and presented in Figure A12 (other in situ sites are not used since they do not provide the SSM measurements). The optimal time lag is less than 1 day between the SSM and 20 cm depth SM at all four RISMA stations, consistent with the vertical coupling between dynamics of satellite SSM and the simulated 0–25 cm SM. Across the four RISMA sites, the optimal time differences between the variations of SSM and the 50 cm SM range from 0 to 5 days (0 day for ON2 and ON6, 1 day for ON5, and 5 days for ON4), which is also comparable to the response time difference (about 2 days in the RISMA region) between satellite SSM and the simulated 25–50 cm SM.***

***The $T_{opt}$ values for SWI estimation based upon the point scale in situ soil moisture measurements at the four RISMA stations are given in Figure A13. The point-scale $T_{opt}$ values range from 1 to 12 days (1 day for ON2, 2 days for ON6, 3 days for ON4, and 12 days for ON5) for SWI estimation at 20 cm depth, while the point-scale $T_{opt}$ values for SWI estimation at 50 cm depth are mostly shorter than 12 days (although the ON4 site shows an $T_{opt}$ of about 50 days for SWI estimation at 50 cm depth, the confidence interval for the $T_{opt}$ is expected to be relatively wide since the highest Spearman's rank correlation varies little over a wide range of T values). Overall, the point-scale $T_{opt}$ values are shorter than those derived from the satellite and model simulated SM for the 9-km grid scale and the watershed-scale. This may indicate that the deeper subsurface layers typically show a quicker response to the near-surface moisture content variability at the point-scale.***”

Some specific comments were provided as below:

1. Evaluation using the in-situ data is an important part of the study. So I suggest moving the Table A1 into the main text.

**Response: The table is now moved to the main text in the revised manuscript** (new Table 1, Page 7)**.**

2. Figs. A1 & A2: There is a mismatch between the model soil layers and in-situ SM observations. Please specify the soil depth of the in-situ data used for the comparison

**Response: The soil depth matching between the in situ SM and model simulations for the two soil profiles (0-25 cm and 0 -100 cm) were described** at Line 185 – 192**:**

*0–25 cm: the simulated SM in the model's top soil layer (0 -25 cm) against a depth-weighted average of in situ measurements in the top 25 cm soil (i.e., 5 cm and 25 cm depths at the RISMA sites, 10 and 20 cm depths at Metcalfe and Pleasant Valley, and 20 cm depth at Winchester stations, see Table 1)*

*0–100 cm: a depth-weighted average of simulated SM from the model's three soil layers (0-25 cm, 25 – 50cm, and 50-100 cm) versus a depth weighted average of in situ measurements in the top 100 cm soil (i.e., 5, 20, and 50 cm depths at the RISMA sites; 10, 20, and 50 cm depths at Pleasant Valley, and 15 and 45 cm depths at WEBS stations, see Table 1)*

.

3. Fig A7: First I suggest the authors using different legend for the layers at top 50cm, and below 50cm. And what explains the spatial heterogeneity in the optimal time lags? It would be good to provide more specific details on this, rather than a general discussion as shown in Lines 315-318.

**Response: The same legend is used for the different soil layers since it can facilitate the time lag comparison/variation across the different depths. In particular, it can clearly show that the optimal time lag increased with the soil depth.**

**The soil drainage properties have a key impact on the spatial heterogeneity in the optimal time lag. In the revision, the averaged optimal time lags are calculated for different soils over the study watershed** (new Tables A1, page 38)**. The relevant discussion is also added:**

Line 326-334 : *"Table A1 provides the average optimal time lags for the six major soils over the study watershed. For each soil, the averaged optimal time lag is calculated using the 9-km SMAP grids dominated by the soil texture (the Organic and Morrisburg soils are not calculated and included in the table due to their insufficient sample grids). Clearly, the soil drainage has a key impact on the spatial variability of the time lags for deeper layers. The optimal time lag for the 25–50 cm depth is statistically shorter (longer) than 1 day in regions with well drained (imperfectly or poorly drained) soils. Moving to the 50–100 cm depth, on average, the soils of*

*Achigan (imperfectly drained) and Bearbrook (poorly drained) dominated regions experienced the longest optimal time delay (close to or higher than 10 days). Further, the optimal time delay is statistically less (more) than 10 days for the GW system in the areas with good (poor or imperfect) soil drainages.*"

Moreover, even though the in-situ data were very scarce, I would like to see a comparison of the optimal time lags derived from in-situ soil moisture data with the values derived from model simulations and SMAP surface SM. Are they comparable?

**Response: In this revision, we have examined the time lags between the in situ SSM (top 5 cm) and the in situ SM at 25 cm/50 cm depths at the four RISMA sites (Figure A12). The relevant discussion is added at Line 568-576:**

*"With the in situ soil moisture measurements at the four RISMA stations, the time lags between the variations of SSM (top 5 cm) and subsurface SM at the point scale are investigated and presented in Figure A12 (other in situ sites are not used since they do not provide the SSM measurements). The optimal time lag is less than 1 day between the SSM and 20 cm depth SM at all four RISMA stations, consistent with the vertical coupling between dynamics of satellite SSM and the simulated 0–25 cm SM. Across the four RISMA sites, the optimal time differences between the variations of SSM and the 50 cm SM range from 0 to 5 days (0 day for ON2 and ON6, 1 day for ON5, and 5 days for ON4), which is also comparable to the response time difference (about 2 days in the RISMA region) between satellite SSM and the simulated 25–50 cm SM."*

4. Fig. 7. The SMAP surface SM seems showing an early thaw onset compared to the model simulations. Why? Does this affect the above time lag analysis?

**Response: When comparing the SMAP near-surface SM (SSM) and the model SSM (Fig. 6), there is no significant difference between them in terms of the thaw onset. In figure 7, the SMAP SSM (top 5 cm) indicates a slightly earlier thaw onset than do the model simulated water content in deeper zones (0–25 cm, 25–50 cm, and 50–100 cm SM). This is not surprising since this reflects a downward heat transfer and migration of thawing front. During a thawing/warming period, the soils typically have a downward temperature gradient (i.e., soil temperature decreases with increased soil depth), which causes a downward heat transfer and migration of thawing front. The thaw onset difference between different depths is consistent with (rather than against) the time lag analysis and the response time differences between satellite SSM and the subsurface water. This issue is now further clarified in the revised manuscript (Line 383-388):**

*"In Fig. 7a, the SMAP SSM (top 5 cm) indicated a slightly earlier thaw onset than the model simulated SM in deeper layers. This reflects a downward heat transfer and migration of thawing front. During a thawing/warming period, the soils typically have a downward temperature gradient (i.e., soil temperature decreases with increased soil depth), which causes a downward*

*heat transfer and migration of thawing front. The thaw onset difference between different depths is consistent with the response time differences between satellite SSM and the subsurface water".*

5. The characteristic time length T: how does this relate to the time lag shown in Section 4? I believe the time lag analysis should provide some useful information on this. Otherwise, what is the use of such analysis?

**Response:  Please see our above response to the overall comment. Either the characteristic time length T (for the SWI approach) or the time lag analysis can be independently used to quantify the links between the SMAP/satellite SSM variability and the deeper subsurface water content fluctuations. Since the two time parameters (the time lag and the characteristic time length) are suitable for the two independent/different approaches, their calculations are completely independent of each other, i.e., the characteristic time length does not rely upon the time lag and vice versa.  The use of the two independent approaches would make the conclusions (ie., SMAP/satellite SSM variability is strongly linked to the deeper subsurface water content fluctuations and can be used to predict/infer subsurface SM and groundwater variability) more robust.**

6. For the comparison between SWI and model simulations, why was the middle layer (i.e. 50cm) ignored?

**Response: In Section 5.1,  the classic optimal characteristic time length Topt values (15 days for 0-20/25 cm soil layer and 20 days for 0-100 cm soil layer, as taken from Wagner et al., 1999) are used. As such, the calculated SWI is entirely independent of the model simulations so that we can compare the SWI to the modeled subsurface soil moisture (in Section 5.1). Because the widely-used/classic Topt value for the 0-50 cm soil layer is not very evident (to the best of our knowledge), the 0-50 cm SWI comparison was not considered in Section 5.1. However, this did not impact the comparison between SWI and model simulations. Since the SWI for the 0-25 cm and the 0-100 cm layers show very good agreement with the model simulations, it is expected that the SWI for the 0-50 cm layer is also in good agreement with the model simulations.**

7. Line 483-: how do these results compared to the point-scale analysis using in-situ SM data?

**Response: In this revision, we have examined the time lags between the in situ SSM (top 5 cm) and the in situ SM at 25 cm/50 cm depths at the four RISMA sites (Figure A12). The relevant discussion is added at Line 568-576:**

*"With the in situ soil moisture measurements at the four RISMA stations, the time lags between the variations of SSM (top 5 cm) and subsurface SM at the point scale are*

*investigated and presented in Figure A12 (other in situ sites are not used since they do not provide the SSM measurements). The optimal time lag is less than 1 day between the SSM and 20 cm depth SM at all four RISMA stations, consistent with the vertical coupling between dynamics of satellite SSM and the simulated 0–25 cm SM. Across the four RISMA sites, the optimal time differences between the variations of SSM and the 50 cm SM range from 0 to 5 days (0 day for ON2 and ON6, 1 day for ON5, and 5 days for ON4), which is also comparable to the response time difference (about 2 days in the RISMA region) between satellite SSM and the simulated 25–50 cm SM."*

8. Line 517-518: I did not see any specific analysis on the relations between Topt and soil texture.

**Response: The relevant analysis is now added:**

Table A2 & Line 447-452: "*Table A2 shows the average $T_{opt}$ for the six major soils over the study watershed. For each soil, the averaged $T_{opt}$ is calculated using the 9-km SMAP grids dominated by the soil texture (the soils of Organic and Morrisburg are not calculated and included in the table due to their insufficient sample grids). Clearly, the spatial variability of $T_{opt}$ is strongly related to the soil drainage class. For the three depth intervals: 0–25 cm, 0–50 cm, and 0–100 cm layers, on average, $T_{opt}$ exceeds 20 days, 24 days, and 30 days, respectively, in regions with imperfect or poor soil drainage, while the $T_{opt}$ values are reduced to below 18 days, 21 days and 28 days, respectively, for the well drained soils.*"

**Reply to RC2**

This study conducted a thorough analysis, comparing SMAP soil moisture and the Soil Water Index (SWI) with both in situ measurements and simulations from HydroGeoSphere. The paper is commendably well-written and organized. However, discerning novelty and originality proves challenging, as the primary focus seems to be on the comparison and testing of various T values. I would have expected the paper to explore novel methodologies, see references below, beyond the conventional exponential filter method already utilized for SMAP. A more convincing exploration of alternative approaches would enhance the overall contribution of the study.

Li, M.; Sun, H.; Zhao, R. A Review of Root Zone Soil Moisture Estimation Methods Based on Remote Sensing. Remote Sens. **2023**, 15, 5361. https://doi.org/10.3390/rs15225361

Stefan, V.-G.; Indrio, G.; Escorihuela, M.-J.; Quintana-Seguí, P.; Villar, J.M. High-Resolution SMAP-Derived Root-Zone Soil Moisture Using an Exponential Filter Model Calibrated per Land Cover Type. Remote Sens. **2021**, 13, 1112. https://doi.org/10.3390/rs13061112

**Response: Thanks for the comments. Although the authors agree that exploring other approaches (e.g., data assimilation and machine learning) would be also very useful for**

quantifying the potential of using SMAP soil moisture (SM) variability to predict subsurface water dynamics, the authors feel that the novelty of using a fully-integrated groundwater (GW)-surface water (SW) model for this type of study should be acknowledged and appreciated. Although the analysis methods used for quantifying the connections between satellite/SMAP soi moisture measurements and modelling results in this study have been widely used previously, fully-integrated groundwater (GW)-surface water (SW) models have not yet been used for this type of study. How a fully-integrated GW-SW model deals with soil moisture, GW, and runoff/streamflow is different from land surface models which are widely used previously. As stated in the manuscript (~Line 80-85), fully-integrated GW-SW model present better ability to reproduce realistic root zone SM and GW dynamics than surface-water models used in previous studies. Hence, these models are well suited to help expand our understanding of connections between satellite/SMAP SM and the variably-saturated subsurface flow regime. Therefore, the use of a fully-integrated groundwater (GW)-surface water (SW) model HydroGeoSphere (HGS) and the examination of the connections between SMAP soil moisture and the HGS simulations in this study has provided a novel exploration for this field.  The specified novelty and advances are discussed in Section 6.1 (Line 465-525).

Further, the authors have also been exploring other approaches (e.g., data assimilation and machine learning) for this type of study. Some preliminary results were presented on conferences (e.g.,  X Xu, SK Frey, AK Nayak, Assimilation of SMAP Soil Moisture for Advancing Integrated Groundwater-Surface Water Modeling, AGU Fall Meeting 2022;  AK Nayak, X Xu, SK Frey, Improving Physically-Based Hydrologic Predictions With Deep Learning, AGU Fall Meeting 2023). The detailed analysis results will be presented in separate manuscripts that are in preparation.

---

## Author Response (AR3)

**Author's response to the comments**

**Editor's remarks:**

The authors did not discuss use of the other SMAP soil moisture products for the study. Such as the SMAP-Sentinel 3km/1km soil moisture product and the Level-4 soil moisture product. Especially, how the study performance is different from the SMAP Level-4 root-zone soil moisture product. Highlight especially why SMAP other products were not considered and how the methods proposed in the study is apart from the SMAP Level-4 root-zone soil moisture product to assess subsurface water dynamics. Present the comparison of the results when you consider SMAP Level-4 product against your approach.

**Response: Thank you for the remarks. The relevant discussion is now added.**

Page 31: Line 636-650:

**"6.4 Other SMAP soil moisture products**

*In this study, only the SMAP enhanced L3 radiometer 9 km EASE-grid SM (SPL3SMP_E) product (O'Neill et al., 2021) was used. The SMAP/Sentinel-1 L2 Radiometer/Radar SM product (Das et al., 2019; 2020), which can provide higher spatial resolution (3 km and 1 km) SSM, was not used here because the temporal resolution of the product (~ 12 days) is not appropriate for detecting the time lags between the variations of SSM and subsurface SM. Further, although the SMAP Level-4 (L4) product can provide the surface (0-5 cm) and root-zone (0-100 cm) SM data at 3-h intervals over 9-km EASE-grid (Reichle, et al., 2022), the product is also not suitable for the approaches utilized in this study since the L4 root-zone SM variability is not independent of the SMAP L3/L4 SSM variability. The links between the SMAP SSM and L4 root -zone SM variations are controlled by the Catchment land surface model and the assimilation system of SMAP brightness temperatures that were used for producing the L4 product. However, note that the SMAP L4 product is in very good agreement with the HGS model simulations, which were used for representing the subsurface water dynamics in this work, in terms of the root zone SM variability (Figure 14A; the absolute bias between them has no impact on the approaches used in this work, which considering only the temporal variations of SM). This further supports the HGS model's application towards representing the dynamic behavior of subsurface water in this work.*"